# Improving Feasibility via Fast Autoencoder-based Projections

**Maria Chzhen**
University of Toronto
Toronto, ON, Canada
`maria.chzhen@mail.utoronto.ca`

**Priya L. Donti**
Massachusetts Institute of Technology
Cambridge, MA, USA
`donti@mit.edu`

## Abstract

Enforcing complex (e.g., nonconvex) operational constraints is a critical challenge in real-world learning and control systems. However, existing methods struggle to efficiently enforce general classes of constraints. To address this, we propose a novel data-driven amortized approach that uses a trained autoencoder as an approximate projector to provide fast corrections to infeasible predictions. Specifically, we train an autoencoder using an adversarial objective to learn a structured, convex latent representation of the feasible set. This enables rapid correction of neural network outputs by projecting their associated latent representations onto a simple convex shape before decoding into the original feasible set. We test our approach on a diverse suite of constrained optimization and reinforcement learning problems with challenging nonconvex constraints. Results show that our method effectively enforces constraints at a low computational cost, offering a practical alternative to expensive feasibility correction techniques based on traditional solvers.[1]

## 1 Introduction

Many learning and control systems, across areas such as robotics, energy systems, and industrial automation, must produce outputs that respect difficult-to-satisfy (often nonconvex) constraints. Enforcing these constraints reliably and efficiently is crucial for both safety and real-world usability. A number of approaches have been proposed to address this challenge within learning-based systems, including penalty methods (Fioretto et al., 2021; Stooke et al., 2020), differentiable projection and correction methods (Chen et al., 2021; Pan et al., 2022), and post-hoc repair algorithms (Zamzam & Baker, 2020; Nocedal & Wright, 2006). However, each of these approaches comes with distinct trade-offs in terms of reliability, generality, computational cost, and solution quality – for instance, failing to satisfy constraints in practice, not handling general classes of constraints, being slower than desired (and potentially prohibitively slow) to deploy in real-world systems, and/or degrading end-to-end model performance.

To address this challenge, we propose a data-driven amortized alternative to traditional constraint enforcement algorithms: a trained autoencoder that acts as an approximate projector to provide fast corrections to infeasible predictions. Specifically, we train an autoencoder to learn a structured, convex latent representation of the feasible set, in a way that enables rapid mapping from infeasible to feasible points. This autoencoder can then be leveraged as a plug-and-play "attachment" to standard neural networks. While not aiming to supersede strict projections in regimes demanding hard feasibility, this approach is designed to improve feasibility in low-latency settings or in settings where moving predictions closer to the feasible set suffices to guarantee downstream system performance. Our key contributions are as follows:

- **Framework for data-driven feasibility improvement.** We pose an approach for *learning* feasibility improvement mappings that can be appended to neural networks. We propose a particular instantiation of this approach, FAB, which employs an autoencoder as the basis of

---

[1]Code for the method and all experiments can be found at: `https://github.com/MOSSLab-MIT/Fast-Autoencoder-Based-Projections`

the learned mapping. Our approach is useful for learning topologically difficult constraints which may not have analytical solvers, as well as providing an alternative to expensive (albeit exact) constraint enforcement approaches.

- **Structured latent representation learning.** We propose a mechanism to learn faithful, feasibility-preserving latent representations of the feasible set, via adversarial training.

- **Empirical validation.** We test our method on a range of constrained optimization and reinforcement learning (RL) problems. In our constrained optimization settings, the method consistently provides an efficient approximate projection, learning to map solutions to the feasible set close to 100% of the time in a fraction of a millisecond (faster than any other method). For RL settings, the method consistently provides safer actions than methods like proximal policy optimization (PPO) and trust-region policy optimization (TRPO), as well as their constrained variants. Overall, this demonstrates the initial promise of our approach in providing fast, reliable feasibility across a wide range of settings.

## 2 RELATED WORK

**Amortized optimization with constraints.** Also known as *learning to optimize*, amortized optimization is a paradigm designed to accelerate the process of solving optimization problems by using machine learning models as fast function approximators (Amos, 2023). Models are trained either via supervised learning on a dataset of known solutions (Chen et al., 2022), or through unsupervised/self-supervised methods, where the model's loss function incorporates the optimization problem's objectives and constraints (Van Hentenryck, 2025). A primary challenge in amortized optimization is ensuring that the model's output satisfies constraints. Several lines of work have emerged to address this, including penalty methods, differentiable projection and/or correction methods, and post-hoc "repair" techniques. Penalty methods turn constrained optimization problems into unconstrained ones by incorporating penalty terms for constraint violations into the optimization objective (Fioretto et al., 2021; Stooke et al., 2020; Raissi et al., 2019); while these methods *incentivize* feasibility, they do not guarantee it, and may produce highly infeasible solutions in practice. In contrast, differentiable projection and/or correction methods embed exact solvers directly as layers in neural networks (Pham et al., 2018; Chen et al., 2021; Pan et al., 2022; Nguyen & Donti, 2025; Donti et al., 2021). While these methods do provide feasibility guarantees, they are often expensive to run, or highly specialized to certain classes of constraints (Min & Azizan, 2024; Tordesillas et al., 2023). Likewise, post-hoc "repair" methods (Boyd et al., 2011; Douglas & Rachford, 1956; Zamzam & Baker, 2020) are often either expensive or highly specialized, and the inherent train-test mismatch can further degrade overall performance.

In this work, we propose an alternative, data-driven approach that can learn an inexpensive, non-iterative transformation from a latent convex set to any arbitrary constraint set, thereby speeding up feasibility improvement in practice (albeit at the expense of provable guarantees). The work closest in the literature to ours is (Liang et al., 2024; 2023), proposing a homeomorphic projection approach which learns an invertible mapping from a closed hyperball to a topologically-equivalent constraint set, and then uses a bisection procedure to recover feasibility. However, this procedure is designed for post-hoc feasibility correction rather than end-to-end training, and is limited to ball-homeomorphic constraint sets. Our work presents a method that can learn a mapping between a latent closed hyperball and any continuous constraint set, and is compatible with end-to-end training and inference. In addition, while homeomorphic projections rely on an iterative bisection procedure, FAB projections are one-shot, leading to significant speed gains in practice.

**Safe reinforcement learning.** Safe reinforcement learning (RL) is an extension of standard RL, where an agent learns to make a sequence of decisions in an environment to maximize a cumulative reward signal while also aiming to minimize the costs associated with constraint violations (García & Fernández, 2015; Gu et al., 2024). While standard RL algorithms, such as Proximal Policy Optimization (PPO, Schulman et al., 2017) and Trust Region Policy Optimization (TRPO, Schulman et al., 2015) excel in unconstrained settings, they are not equipped to handle constraints. Prominent approaches to address this include Lagrangian relaxation (Stooke et al., 2020), safe exploration (Hans et al., 2008), and differentiable projection (Pham et al., 2018; Chen et al., 2021). However, mirroring the discussion on constraint enforcement in amortized optimization, these approaches come with distinct trade-offs in terms of reliability and computational cost.

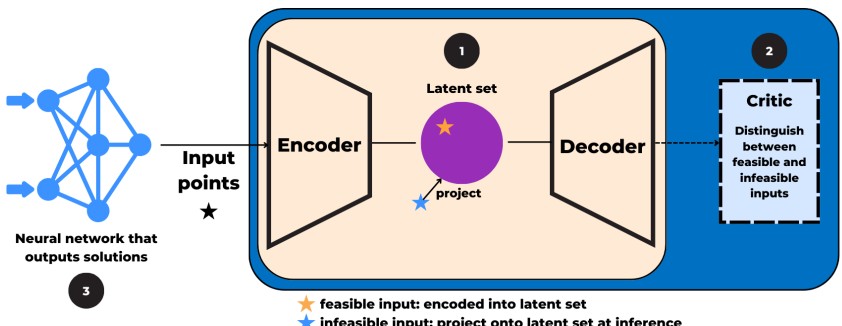

Figure 1: A schematic of FAB approximate projections. (1) Phase 1 of autoencoder training aims to enable reconstructions of the feasible set. (2) Phase 2 of autoencoder training introduces a discriminator (critic) to enable further structuring and refinement of the latent representation. (3) The trained autoencoder can be utilized as a plug-and-play attachment to another neural network model.

**Adversarial training.** Adversarial training, widely known for its application in Generative Adversarial Networks (GANs) (Goodfellow et al., 2014) and adversarially robust deep learning (Bai et al., 2021), offers a general framework for learning in the face of complex data distributions. In the canonical GAN setup, a *generator* network learns to produce realistic data samples from a noise vector, while a *discriminator* is trained to distinguish between real and generated samples. Minimax training of the generator and discriminator drives the generator to produce increasingly plausible outputs. Inspired by this approach, our work uses adversarial training to learn latent representations of feasible sets, where the "discriminator" is trained to distinguish between feasible and infeasible outputs produced by another model, and that output-producing model is thereby forced to learn more robust decision boundaries between feasible and infeasible points.

## 3 FAST AUTOENCODER-BASED (FAB) FEASIBILITY IMPROVEMENT

In this paper, we consider the task of repeatedly solving parametric optimization problems of the form:

$$y^\star(x) \in \arg\min_{y \in \mathcal{C}(x)} f(y; x), \tag{1}$$

where $x \in \mathcal{X} \subseteq \mathbb{R}^m$ are the problem parameters, $y \in \mathcal{Y} \subseteq \mathbb{R}^n$ are decision variables, $f : \mathcal{Y} \times \mathcal{X} \to \mathbb{R}$ is the objective function, $\mathcal{C}(x) \subseteq \mathcal{Y}$ is a (parameter-dependent) inequality constraint set, [2] and $y^\star(x) \in \mathcal{C}(x)$ is a solution. Because the optimal solutions of the problems change as the parameters $x$ change, this suggests a mapping between parameters $x$ and optimal solutions. In addition to offline optimization, this formulation also captures RL settings, where $\mathcal{X}$ and $\mathcal{Y}$ are the state and action spaces, respectively (Amos, 2023). Our aim is to use neural networks to approximate a mapping from $x$ to $y^\star(x)$ such that the resultant solution has low objective value, satisfies constraints, and is fast to compute at inference time.

We specifically consider neural networks of the form:

$$\hat{y}_\theta := \phi_x \circ N_\theta, \tag{2}$$

where $N_\theta : \mathcal{X} \to \mathbb{R}^d$ is a standard feedforward network with parameters $\theta$, and $\phi_x : \mathbb{R}^d \to \mathcal{C}(x)$ is a differentiable feasibility improvement procedure that aims to map to a point in the constraint set. Prior work has considered the design of mappings $\phi_x$ that can ensure exact feasibility, e.g., by solving a differentiable orthogonal projection or via exact iterative procedures (see Section 2). While such approaches are indeed useful for, e.g., safety-critical settings where provable constraint satisfaction is a must, they tend to be relatively expensive for general classes of constraints. Instead, we propose to learn a fast, data-driven, *approximate* mapping for use in settings where (near-)feasibility is important, but where the benefits of reduced latency outweigh the need for strict constraint satisfaction.

---

[2]While our framework is unlikely the best fit to directly handle equality constraints, it could in principle readily be combined with equality constraint "completion" approaches such as those in Donti et al. (2021); Pan et al. (2022).

---

**Algorithm 1** Two-Phase Autoencoder Training for FAB Feasibility Improvement

---

1: **input:** Dataset of feasible points $\mathcal{T}_{\text{feas}} = \{(y^{(i)}, x^{(i)}) \mid y^{(i)} \in \mathcal{C}(x^{(i)})\}$
2: **input:** Discriminator dataset $\mathcal{T}_{\text{disc}} := \{((y^{(i)}, x^{(i)}), c^{(i)}) \mid c^{(i)} = 1 \text{ if } y^{(i)} \in \mathcal{C}(x^{(i)}), 0 \text{ otherwise}\}$
3:
4: **procedure** PHASE1($\mathcal{T}_{\text{feas}}$)                 *// Constraint set reconstruction*
5:      **init** encoder $E_\gamma$, decoder $R_\psi$
6:      **while** not converged **do** for batch of $(y, x) \in \mathcal{T}_{\text{feas}}$
7:          **compute** $\mathcal{L}_{\text{recon}}(y, x)$ via Eq. 4
8:          **update** $\gamma, \psi$ using $\nabla_\gamma \mathcal{L}_{\text{recon}}, \nabla_\psi \mathcal{L}_{\text{recon}}$
9:      **end while**
10:      **return** $E_\gamma$, $R_\psi$
11: **end procedure**
12:
13: **procedure** PHASE2($\mathcal{T}_{\text{disc}}$, $E_\gamma$, $R_\psi$)           *// Latent set structuring*
14:      **init** discriminator $D_\xi$
15:      **while** not converged **do** for batch of $((y, x), c) \in \mathcal{T}_{\text{disc}}$
16:          *// Update discriminator $D_\xi$*
17:          **compute** $\mathcal{L}_{\text{disc}}(y, x, c)$ via Eq. 5
18:          **update** $\xi$ using $\nabla_\xi \mathcal{L}_{\text{disc}}$
19:          *// Update autoencoder $(E_\gamma, R_\psi)$*
20:          **compute** $\mathcal{L}_{\text{recon}}(y, x)$, $\mathcal{L}_{\text{hinge}}(y, x, c)$ via Eq. 4, 7
21:          **sample** batch of $z \sim \mathcal{S}$
22:          **compute** $\mathcal{L}_{\text{latent}}(z)$, $\mathcal{L}_{\text{geom}}$(batch of $z$) via Eq. 8–9
23:          **compute** $\mathcal{L}_{\text{struc}}$ from $\mathcal{L}_{\text{recon}}, \mathcal{L}_{\text{hinge}}, \mathcal{L}_{\text{latent}}, \mathcal{L}_{\text{geom}}$ via Eq. 6
24:          **update** $\gamma, \psi$ using $\nabla_\gamma \mathcal{L}_{\text{struc}}, \nabla_\psi \mathcal{L}_{\text{struc}}$
25:      **end while**
26:      **return** $E_\gamma$, $R_\psi$
27: **end procedure**

---

The particular choice of $\phi_x$ that we propose in this work is motivated by the observation that while $\mathcal{C}(x)$ may be expensive to enforce directly, we can potentially learn a transformation between this set and some other specially-structured set that is much cheaper to work with. In particular, let $E_\gamma : \mathcal{I} \to \mathcal{Z}$ be an encoder with parameters $\gamma$, where $\mathcal{I} := \mathcal{Y} \times \mathcal{X}$ and $\mathcal{Z} \subseteq \mathbb{R}^k$, and let $R_\psi : \mathcal{Z} \to \mathcal{Y}$ be a decoder ("reconstructor") with parameters $\psi$, where both $E_\gamma$ and $R_\psi$ are standard feedforward neural networks.[3] In addition, let $\mathcal{S} \subseteq \mathcal{Z}$ be a set that is by construction simple to map onto (e.g., a simplex or a ball), and let $\phi^{\mathcal{S}} : \mathcal{Z} \to \mathcal{S}$ be a cheap exact mapping to points in $\mathcal{S}$ (e.g., a softmax operation or closed-form projection). We then define our feasibility improvement procedure as

$$\phi_x := R_\psi \circ \phi^{\mathcal{S}} \circ E_\gamma. \tag{3}$$

In other words, given an output from $N_\theta$ alongside its corresponding input, we feed these into an encoder, map the encoded latent point into $\mathcal{S}$, and then decode the resultant point. The idea is that once $E_\gamma$ and $R_\psi$ are trained, each step of $\phi_x$ is cheap to execute, leading to fast overall inference.

The key challenge with this approach is that we must now design the autoencoder parameters $(\gamma, \psi)$ such that the output of $\phi_x$ is actually feasible, i.e., actually lies within $\mathcal{C}(x)$. To do this, we propose a two-stage autoencoder training procedure aimed at structuring the latent space $\mathcal{Z}$ such that, to the extent possible, latent points decode to our feasible set $\mathcal{C}(x)$ if and only if they lie within $\mathcal{S} \subseteq \mathcal{Z}$. A schematic of our overall approach is provided in Figure 1. We now provide additional detail on the two-stage training procedure.

### 3.1 TWO-PHASE AUTOENCODER TRAINING

We train the encoder-decoder pair $(E_\gamma, R_\psi)$ in two phases: (1) a *constraint set reconstruction phase* using standard autoencoder training, and (2) a *latent space structuring phase* involving adversarial

---

[3]We focus our discussion in the main text on the case of input-varying constraints $\mathcal{C}(x)$, but note that the formulations for input-independent constraints are similar. In the latter case, $\mathcal{I} := \mathcal{Y}$, our autoencoder is simply a standard (rather than conditional) autoencoder, and all mentions of $x$ in Section 3.1 can be dropped.

training with a discriminator. Together, these aim to enable the autoencoder to faithfully reconstruct the feasible set, and encourage latent points from $\mathcal{S}$ to decode to feasible points. Pseudocode for both phases is in Algorithm 1.

**Phase 1: Constraint set reconstruction.** We first train the autoencoder to reconstruct the feasible set. Specifically, we train on a dataset of exclusively feasible points, $\mathcal{T}_{\text{feas}} := \{(y^{(i)}, x^{(i)}) \mid y^{(i)} \in \mathcal{C}(x^{(i)})\}$. As standard in autoencoder training, the encoder $E_\gamma$ and the decoder $R_\psi$ are jointly trained to minimize a standard $L_2$ reconstruction loss to prioritize the fidelity of decoded points:

$$\mathcal{L}_{\text{recon}}(y, x) = \|y - R_\psi(E_\gamma(y, x))\|_2^2. \tag{4}$$

**Phase 2: Latent space structuring.** The second autoencoder training phase aims to structure the latent space such that any point sampled from $\mathcal{S} \subseteq \mathcal{Z}$, when decoded, results in a feasible point. To do this, we draw loose inspiration from generative adversarial network (GAN) training, and train our autoencoder alternatingly with a discriminator $D_\xi : \mathcal{I} \to [0, 1]$ whose role is to distinguish between feasible and infeasible points. Specifically, the discriminator maps from inputs in $\mathcal{I}$ to an estimated probability that the input is feasible. We let $D_\xi$ be a standard feedforward neural network.

We leverage two datasets during this training phase: (a) a labeled dataset of feasible *and* infeasible points, $\mathcal{T}_{\text{disc}} := \left\{ \left( (y^{(i)}, x^{(i)}), c^{(i)} \right) \mid c^{(i)} = 1 \text{ if } y^{(i)} \in \mathcal{C}(x^{(i)}), 0 \text{ otherwise} \right\}$, and (b) samples $z^{(j)} \sim \mathcal{S}$ from our latent subset. Unlike in GANs, where the generator and discriminator are trained via a minimax loss, here, the autoencoder and discriminator are trained using distinct loss functions. Specifically, the discriminator $D_\xi$ is trained to distinguish between feasible and infeasible points, by minimizing the negative log likelihood on $\mathcal{T}_{\text{disc}}$:

$$\mathcal{L}_{\text{disc}}(y, x, c) = - \left( c \log D_\xi(y, x) + (1 - c) \log \left( 1 - D_\xi(y, x) \right) \right). \tag{5}$$

The autoencoder is trained to minimize a composite loss function using both $\mathcal{T}_{\text{disc}}$ and $z^{(j)} \sim \mathcal{S}$, defined as:

$$\mathcal{L}_{\text{struc}} = \frac{1}{N} \sum_{i=1}^{N} \left[ \lambda_{\text{recon}} \mathcal{L}_{\text{recon}}(y^{(i)}, x^{(i)}) \right] + \frac{1}{N} \sum_{i=1}^{N} \left[ \lambda_{\text{hinge}} \mathcal{L}_{\text{hinge}}(y^{(i)}, x^{(i)}, c^{(i)}) \right] \\ + \frac{1}{M} \sum_{j=1}^{M} \left[ \lambda_{\text{latent}} \mathcal{L}_{\text{latent}}(z^{(j)}) \right] + \lambda_{\text{geom}} \mathcal{L}_{\text{geom}}(\{z^{(1)}, \ldots, z^{(M)}\}), \tag{6}$$

where $\lambda_{\text{recon}}, \lambda_{\text{latent}}, \lambda_{\text{hinge}}, \lambda_{\text{geom}} \in \mathbb{R}$; $\mathcal{L}_{\text{recon}}$ is defined above; and the remaining loss terms are defined as:

- The hinge loss $\mathcal{L}_{\text{hinge}}$: Structures the latent space by mapping feasible points to the closed hyperball and infeasible points to outside of it. Here, we write the hinge loss for the specific choice of $\mathcal{S} := \{z : \|z\|_2 \le r\}$ for some radius $r$:

$$\mathcal{L}_{\text{hinge}}(y, x, c) = c \, \text{ReLU} \left( \|E_\gamma(y, x)\|_2 - r \right) + (1 - c) \, \text{ReLU} \left( r - \|E_\gamma(y, x)\|_2 \right). \tag{7}$$

- The latent loss $\mathcal{L}_{\text{latent}}$: Encourages outputs decoded from latent points $z \in \mathcal{S}$ to be feasible, via supervision from the discriminator:

$$\mathcal{L}_{\text{latent}}(z) = - \log D_\xi(R_\psi(z)). \tag{8}$$

- The Jacobian regularization term $\mathcal{L}_{\text{geom}}$: Encourages uniform coverage of the feasible set by the decoder (Nazari et al., 2023), and is computed over a set of latent points $\hat{\mathcal{S}} \subseteq \mathcal{S}$:

$$\mathcal{L}_{\text{geom}}(\hat{\mathcal{S}}) = \text{Variance}_{z \sim \hat{\mathcal{S}}} \left[ \log \det(J_z J_z^\top + \varepsilon I_k) \right], \tag{9}$$

where $J_z = \nabla_z R_\psi(z)$ is the Jacobian of the decoder with respect to $z$, $I_k$ is the $k \times k$ identity matrix, and $\varepsilon$ is a small scalar value. This loss term measures how the decoder's output changes under small changes in the latent code ($\varepsilon I_k$). Specifically, it calculates the determinant of the Gram matrix $J_z J_z^\top$ and measures how much the decoder locally stretches or shrinks the space.

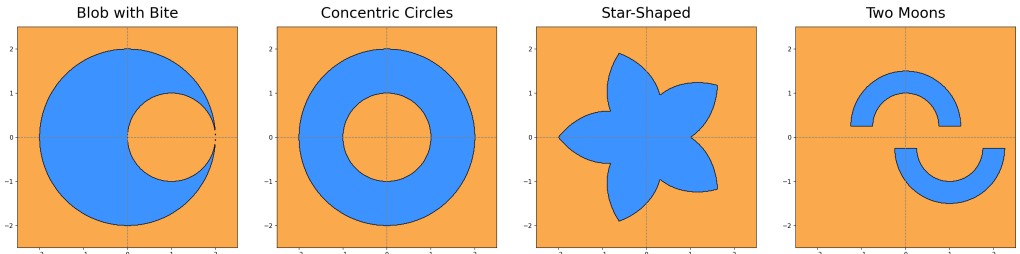

Figure 2: The nonconvex constraint sets tested in our constrained optimization settings, termed (from left to right): Blob with Bite, Concentric Circles, Star-Shaped, and Two Moons.

### 3.2 LEVERAGING THE TRAINED FEASIBILITY MAPPING

After autoencoder training, we fix the weights of the encoder and decoder and use them to construct the feasibility mapping $\phi_x = R_\psi \circ \phi^{\mathcal{S}} \circ E_\gamma$, as given by Equation 3. We then append the feasibility mapping to our base neural network as $\hat{y}_\theta = \phi_x \circ N_\theta$, as given by Equation 2. The resultant neural network $\hat{y}_\theta$ (with learnable parameters $\theta$) can then be trained end-to-end as usual to address the amortized optimization problem at hand, i.e., to aim to learn optimal solutions to Problem 1. In particular, $N_\theta$ learns to adapt to the autoencoder's mapping procedure as part of end-to-end training, i.e., can account for this mapping in picking neural network parameters aimed at improving the optimality of the end-to-end solution. (While the encoder/decoder parameters $\psi, \gamma$ could also be further trained end-to-end alongside the parameters $\theta$, rather than being fixed, we do not consider that case here.) Overall, our learned mapping serves as a plug-and-play attachment to standard deep learning models that can be used during training and inference to improve model feasibility, while also being fast to run. In the following sections, we demonstrate the performance of $\hat{y}_\theta$ in learning feasible amortized optimization solutions in both offline optimization and RL settings.

## 4 EXPERIMENTS ON CONSTRAINED OPTIMIZATION PROBLEMS

**Problem classes**. We test our approach on constrained optimization problems with three objective types: linear, quadratic, and distance minimization. We consider four classes of 2-dimensional nonconvex constraint families (Figure 2) as well as 3-, 5-, and 10-dimensional hyperball constraints $\mathcal{C} := \{x \in \mathbb{R}^d \mid 1 \leq \|x\|_2^2 \leq 2\}$. We note that these are input-independent constraint sets $\mathcal{C} \subseteq \mathcal{Y}$ (we will consider input-dependent constraints in Section 5) that do not include equality constraints. The problems are formulated as:

Linear:
$$\min_y a^\top y$$
$$\text{s.t. } y \in \mathcal{C}$$

Quadratic:
$$\min_y y^\top Q y + a^\top y$$
$$\text{s.t. } y \in \mathcal{C}$$

Distance minimization:
$$\min_y \left\| y - t \right\|^2$$
$$\text{s.t. } y \in \mathcal{C},$$

where $y \in \mathcal{Y} := \mathbb{R}^n$ is the decision variable, and problem parameters $a \in \mathbb{R}^n, Q \in \mathbb{R}^{n \times n}, t \in \mathbb{R}^n$ vary between instances. Our goal is to learn a neural network approximator $\hat{y}_\theta$ for each problem class that minimizes the objective while satisfying constraints.

**Baselines**. We compare FAB feasibility improvement against:

- **Projected Gradient Descent**. A classical optimization algorithm that, starting from an initial point $y$, takes a gradient step on the objective $f(y)$ and then (approximately) projects to the feasible set by replacing any infeasible iterate with the nearest sampled feasible point.

- **Penalty**. A classical optimization algorithm that solves an unconstrained version of the problem, $\min_y f(y) + \mu v(y)$, where $\mu > 0$ is a penalty weight and $v(y)$ is a differentiable surrogate of constraint violation. The procedure for obtaining $v(y)$ is described in Appendix A.3.

- **Augmented-Lagrangian**. A classical optimization method that combines penalty terms and Lagrange multipliers to handle constraints. The objective is to minimize $f(y) + \lambda v(y) +$

$\frac{\rho}{2}v(y)^2$, where $\lambda$ is the dual variable, $\rho > 0$ is a penalty parameter, and $v(y)$ is the same violation surrogate used by Penalty. The method alternates between gradient-based updates of $y$ and updates of $\lambda$.

- **Interior Point**. A classical optimization algorithm that ensures feasibility by penalizing the optimizer as it approaches constraint boundaries. The objective is $\min f(y) - \mu \log\big(\max\{\tau - v(y), \varepsilon\}\big)$, where $\tau > 0$ is a slack threshold, $\varepsilon > 0$ is for numerical stability, and $\mu$ is annealed over time.

- **Penalty Neural Network** that learns to minimize $f(y) + \mu v(y)$, where $\mu > 0$ is a penalty weight.

- **FSNet** (Nguyen & Donti, 2025). A neural network that predicts a candidate minimizer $y_{\text{pred}}$, refined by an iterative solver minimizing $v(y)$ to produce a feasible $y_{\text{feas}}$. The neural network loss function is $f(y_{\text{feas}}) + \frac{\rho}{2}\|y_{\text{feas}} - y_{\text{pred}}\|^2$, $\rho > 0$.

- **Homeomorphic Projection** (Liang & Chen, 2025). An invertible neural network $\phi$ trained to map a simple latent set onto the feasible region with low geometric distortion. At inference, an input $y$ is inverted to $z = \phi^{-1}(y)$, and bisection along the latent ray returns a feasible output.

**Ablations**. We also test the following ablations and variants of our method:

- **Different numbers of decoder networks:** We examine the effect of training the autoencoder with multiple decoder networks, in principle allowing different decoder networks to "specialize" in different parts of the latent space. Specifically, we consider a collection of $\rho$ decoder networks $R_{\psi_1}^{(1)}, \ldots, R_{\psi_\rho}^{(\rho)}$ each mapping from $\mathcal{Z} \to \mathcal{Y}$. In addition, we introduce a weighting network $M_\omega : \mathcal{Z} \to \mathbb{R}^\rho$ that maps from the latent space to a set of logits (one per decoder). For any $z \in \mathcal{Z}$, we compute mixture weights $\alpha(z) = \text{softmax}(M_\omega(z)) \in \mathbb{R}^\rho$ and define the final decoded output as the convex combination $R_\psi(z) = \sum_{i=1}^\rho \alpha_i(z) R_{\psi_i}^{(i)}$. The weighting network $M_\omega$ and all decoder networks $R_{\psi_i}^{(i)}$ are trained jointly end-to-end using the same loss functions as described in Section 3.1, with $R_\psi(z)$ as the decoded output. In our ablations, we test the settings where $\rho = 2, 3$.

- **Phase 1 Only:** The autoencoder was trained using only the reconstruction loss and no Phase 2.

- **No Hinge Loss:** The autoencoder was trained without the hinge loss.

- **No discriminator** $D_\xi$**:** The adversarial discriminator was removed from Phase 2 training entirely, relying only on the other losses, to isolate the discriminator's contribution.

In Appendix C, we also provide experiments testing the effect of (a) incomplete coverage of the constraint set in the autoencoder training data, and (b) using different decoder depth and width configurations.

**Implementation details**. We run experiments over 5 seeds with 300 problems per seed (trained for 500 epochs with a batch size of 32, tested on 1,500 problems), totaling 18,000 optimization problems across the entire testing suite. All methods were run on a workstation equipped with one NVIDIA RTX 4090 GPU. We choose $\mathcal{S}$ to be a 0.5-radius closed hyperball. Hyperparameters are given in Appendix A.1.

**Evaluation metrics**. We evaluate the methods by their feasibility rates, optimality gaps, and inference times over the 4 constraint types and 3 objective types.

**Results**. As presented in Table 1 and Appendix B, our methods achieve a substantial reduction in computation time, with sub-millisecond inference times (1-2 orders of magnitude faster than exact feasibility methods), while consistently achieving high feasibility rates across the feasible sets. Across most experiments, the best-performing method was found to be FAB with one decoder; FAB with multiple decoders emerges as the strongest method for the Two Moons disjoint constraint set. Notably, we see that FAB does well at finding feasible solutions even in difficult cases with disjoint or non-ball-homeomorphic sets.

| Method | | Quadratic | | | Linear | | | Dist. Min. | | |
|---|---|---|---|---|---|---|---|---|---|---|
| | | ↑ Feas (%) | ↓ Time (ms) | ↓ Gap | ↑ Feas (%) | ↓ Time (ms) | ↓ Gap | ↑ Feas (%) | ↓ Time (ms) | ↓ Gap |
| **Classical** | Projected Gradient | 80.5 | 79.07 | 1.18 | 76.3 | 48.23 | 0.90 | 79.2 | 69.79 | 4.65 |
| | | $\sigma= 39.6$ | $\sigma= 91.65$ | $\sigma= 1.58$ | $\sigma= 42.5$ | $\sigma= 52.86$ | $\sigma= 0.99$ | $\sigma= 40.6$ | $\sigma= 25.09$ | $\sigma= 5.87$ |
| | Penalty Method | 20.4 | 78.39 | 1.36 | 10.4 | 61.58 | 1.52 | 14.6 | 38.67 | 6.39 |
| | | $\sigma= 40.3$ | $\sigma= 71.27$ | $\sigma= 1.67$ | $\sigma= 30.5$ | $\sigma= 111.21$ | $\sigma= 1.49$ | $\sigma= 35.3$ | $\sigma= 10.02$ | $\sigma= 7.07$ |
| | Augmented Lagrangian | 20.9 | 85.71 | 1.33 | 10.9 | 56.98 | 1.56 | 15.7 | 42.58 | 6.43 |
| | | $\sigma= 40.7$ | $\sigma= 119.17$ | $\sigma= 1.59$ | $\sigma= 31.1$ | $\sigma= 62.66$ | $\sigma= 1.56$ | $\sigma= 36.3$ | $\sigma= 6.69$ | $\sigma= 6.70$ |
| | Interior Point | 78.3 | 82.80 | 1.64 | 77.3 | 46.37 | 1.03 | 79.9 | 42.00 | 6.12 |
| | | $\sigma= 41.2$ | $\sigma= 110.99$ | $\sigma= 2.02$ | $\sigma= 41.9$ | $\sigma= 33.68$ | $\sigma= 1.03$ | $\sigma= 40.0$ | $\sigma= 12.03$ | $\sigma= 6.41$ |
| **ML-Based** | Penalty NN | 91.1 | 0.20 | 0.61 | 80.0 | 0.14 | 0.85 | 40.7 | 0.24 | 4.34 |
| | | $\sigma= 28.4$ | $\sigma= 1.02$ | $\sigma= 1.06$ | $\sigma= 38.4$ | $\sigma= 0.04$ | $\sigma= 1.25$ | $\sigma= 49.1$ | $\sigma= 1.30$ | $\sigma= 6.30$ |
| | FSNet | 98.7 | 4.42 | 1.13 | 98.3 | 3.40 | 0.96 | 99.3 | 4.03 | 5.30 |
| | | $\sigma= 11.5$ | $\sigma= 2.77$ | $\sigma= 1.45$ | $\sigma= 12.8$ | $\sigma= 2.08$ | $\sigma= 1.01$ | $\sigma= 8.5$ | $\sigma= 1.67$ | $\sigma= 5.51$ |
| | Homeomorphic Projection | 73.9 | 247 | 1.74 | 76.1 | 240 | 1.67 | 76.1 | 243 | 8.12 |
| | | $\sigma= 43.9$ | $\sigma= 105$ | $\sigma= 1.90$ | $\sigma= 42.6$ | $\sigma= 104$ | $\sigma= 1.48$ | $\sigma= 42.6$ | $\sigma= 105$ | $\sigma= 7.80$ |
| | FAB | 96.8 | **0.55** | **0.85** | 83.5 | **0.43** | 0.73 | 88.3 | **0.41** | 4.22 |
| | | $\sigma= 17.6$ | $\sigma= 0.40$ | $\sigma= 1.16$ | $\sigma= 37.1$ | $\sigma= 0.27$ | $\sigma= 0.73$ | $\sigma= 32.2$ | $\sigma= 0.20$ | $\sigma= 4.15$ |
| | FAB: 2 Decoder Nets | **100.0** | 0.59 | 1.31 | **100.0** | 0.55 | 0.94 | **100.0** | 0.50 | 5.64 |
| | | $\sigma= 0.0$ | $\sigma= 0.35$ | $\sigma= 1.36$ | $\sigma= 0.0$ | $\sigma= 0.37$ | $\sigma= 0.89$ | $\sigma= 0.0$ | $\sigma= 0.25$ | $\sigma= 5.73$ |
| | FAB: 3 Decoder Nets | **100.0** | 0.80 | 1.02 | **100.0** | 0.62 | 0.90 | **100.0** | 0.60 | 5.44 |
| | | $\sigma= 0.0$ | $\sigma= 1.03$ | $\sigma= 1.15$ | $\sigma= 0.0$ | $\sigma= 0.36$ | $\sigma= 0.84$ | $\sigma= 0.0$ | $\sigma= 0.32$ | $\sigma= 4.84$ |
| | FAB: Phase 1 Only | 82.5 | 0.64 | 0.86 | 69.9 | 1.07 | 0.72 | 64.1 | 0.54 | 3.83 |
| | | $\sigma= 38.0$ | $\sigma= 0.14$ | $\sigma= 1.21$ | $\sigma= 45.9$ | $\sigma= 4.12$ | $\sigma= 0.69$ | $\sigma= 48.0$ | $\sigma= 0.09$ | $\sigma= 4.07$ |
| | FAB: No Hinge Loss | 28.4 | 0.63 | 1.13 | 18.8 | 0.76 | 0.85 | 28.1 | 0.56 | 5.14 |
| | | $\sigma= 45.1$ | $\sigma= 0.16$ | $\sigma= 1.23$ | $\sigma= 39.1$ | $\sigma= 0.84$ | $\sigma= 0.80$ | $\sigma= 44.9$ | $\sigma= 0.16$ | $\sigma= 4.96$ |
| | FAB: No $D_\xi$ | 75.1 | 0.64 | 0.86 | 54.4 | 0.67 | **0.70** | 13.6 | 0.55 | **3.61** |
| | | $\sigma= 43.2$ | $\sigma= 0.22$ | $\sigma= 1.21$ | $\sigma= 49.8$ | $\sigma= 0.31$ | $\sigma= 0.69$ | $\sigma= 34.3$ | $\sigma= 0.19$ | $\sigma= 4.00$ |

Table 1: Mean and std. dev. for feasibility (%), time (ms), and optimality gap for each method: Two Moons.

## 5 EXPERIMENTS ON SAFETY GYM (SAFE RL)

**Safe RL problem formulation**. Safe RL can be framed within our parametric optimization setup from Equation 1. We consider a discrete-time dynamical system where the state evolves according to:

$$s_{k+1} = f(s_k, u_k), \tag{10}$$

where $s_k \in \mathcal{X} \subseteq \mathbb{R}^m$ is the current state and $u_k \in \mathcal{Y} \subseteq \mathbb{R}^n$ is the control action at timestep $k$. In this context, the actions $u_k$ are the decision variables, and the states $s_k$ correspond to the problem parameters. The constraint set $\mathcal{C}(s_k)$ is state-dependent (i.e., problem-parameter-dependent). An action $u_k$ is considered safe, i.e., $u_k \in \mathcal{C}(s_k)$, if executing it from state $s_k$ does not lead to an immediate constraint violation (e.g., collision). The objective is to learn a policy $\pi_\theta : \mathcal{X} \to \mathcal{Y}$, that maximizes the expected cumulative reward while also satisfying constraints.

**Environment**. We evaluate our approach using the SafetyPointGoal2-v0 and SafetyPointPush2-v0 environments from the Safety Gymnasium benchmark suite (Ji et al., 2023). In the Goal task, an agent must navigate to a series of goal locations while avoiding randomly-placed hazards. In the Push task, an agent must navigate to and manipulate a movable box, pushing it toward a target location while avoiding randomly placed hazards. The state vector comprises simulated sensor readings (e.g., accelerometers, gyroscope, and a LiDAR-like sensor for hazard detection), while the action vector controls the agent's forward/backward movement, as well as its rotational velocities.

**Baselines**. We compare the following against the approach of using PPO with FAB projections (**PPO-FAB**):

- **PPO** (Schulman et al., 2017). A state-of-the-art unconstrained RL algorithm.

- **TRPO** (Schulman et al., 2015). Another robust unconstrained RL algorithm.

| Algorithm | Reward Mean ↑ | Reward Std ↓ | Cost Mean ↓ | Cost Std ↓ | Time Mean (s) ↓ |
|---|---|---|---|---|---|
| PPO-FAB | -1.12 | **1.11** | **24.32** | **37.84** | 2.75 |
| PPO | 13.26 | 14.05 | 167.46 | 87.06 | 2.47 |
| PPO-LAG | 2.24 | 5.10 | 54.10 | 64.50 | **2.40** |
| TRPO | **15.58** | 10.31 | 164.14 | 88.43 | 2.59 |
| TRPO-LAG | 2.37 | 8.46 | 89.04 | 187.67 | 2.78 |

Table 2: Results for SafetyPointGoal2-v0 environment.

| Algorithm | Reward Mean ↑ | Reward Std ↓ | Cost Mean ↓ | Cost Std ↓ | Time Mean (s) ↓ |
|---|---|---|---|---|---|
| PPO-FAB | -0.07 | **0.52** | **7.70** | **37.51** | **3.71** |
| PPO | 0.41 | 3.11 | 59.86 | 120.18 | 3.78 |
| PPO-LAG | **0.60** | 1.58 | 31.34 | 58.17 | 4.09 |
| TRPO | 0.20 | 2.50 | 106.88 | 216.19 | 4.03 |
| TRPO-LAG | -0.77 | 5.51 | 28.22 | 67.21 | 4.05 |

Table 3: Results for SafetyPointPush2-v0 environment.

- **Lagrangian PPO (PPO-LAG)** and **Lagrangian TRPO (TRPO-LAG)** (Stooke et al., 2020; Ray et al., 2019). Constrained versions of PPO and TRPO that use Lagrangian relaxation to penalize constraint violations during training.

**Implementation details**. For each environment, the FAB autoencoder was trained on a dataset of 100,000 state-action pairs, approximately evenly balanced between safe and unsafe examples, which were collected by running a random policy in the environment. The performance of all algorithms was evaluated over 50 independent seeds. All methods were run on a workstation equipped with one NVIDIA RTX 4090 GPU. We choose $\mathcal{S}$ to be a 0.5-radius closed hyperball. Hyperparameters are given in Appendix A.2.

**Evaluation metrics**. We assess performance based on several metrics, averaged over the evaluation episodes: episode rewards, episode costs, and mean inference time. Episode reward is the cumulative rewards obtained by the agent in one episode (which is 1000 steps). Episode cost is the total number of constraint violations (i.e., entering hazard zones).

**Results**. As presented in Tables 2 and 3, PPO-FAB achieves the lowest constraint violation costs among the evaluated methods, substantially outperforming standard baselines like PPO and TRPO on this metric, and even somewhat outperforming the safe variants PPO-LAG and TRPO-LAG. Furthermore, it also exhibits the highest levels of reliability, in the sense of having the lowest standard deviations in both cost and reward across all methods compared. However, this focus on safety also corresponds to a more conservative reward-seeking policy, resulting in lower reward means (sometimes much lower) compared to the other algorithms.

## 6 CONCLUSION

In this work, we introduced a novel-data driven method, Fast Autoencoder-Based (FAB) projections, to address the challenge of efficiently enforcing nonconvex constraints in learning-based systems. Our method specially trains autoencoders to learn approximate projections onto general (e.g., potentially nonconvex or disjoint) feasible sets. The trained autoencoders can be leveraged as plug-and-play attachments to standard neural networks as part of a broader constrained learning pipeline. By structuring the autoencoder's latent space to correspond to a simple convex shape that is easy to map into, we can perform a fast projection within this latent space at inference and decode the result back into a point that is highly likely to be feasible.

On our constrained optimization benchmarks, we show FAB consistently achieves near-perfect feasibility rates (often approaching 100%) across a diverse set of challenging, nonconvex feasible sets. In the SafetyGym benchmark, it consistently achieves low constraint violation costs. Crucially, it achieves both of these with inference times that are significantly faster than those of other methods. While methods with formal guarantees can ensure exact feasibility, their computational overhead can

be prohibitive for certain real-time applications. Therefore, FAB presents a compelling alternative for latency-sensitive domains where rapid, high-fidelity approximate projections may be sufficient.

Limitations of FAB include lack of hard guarantees, and its reliance on having a representative dataset of feasible and infeasible points in order for autoencoder training to perform well. We also observe that the adversarial training process can be somewhat unstable, requiring, e.g., extensive hyperparameter tuning. We also note that the experiments in this work, while spanning diverse settings, are relatively small-scale. Future work includes both larger-scale evaluations and identifying ways to improve sample efficiency, distributional performance, and interpretability of the data-driven mapping – e.g., through the use of specialized neural network structures such as input-convex neural networks (Amos et al., 2017) or operator learning-based networks (Kovachki et al., 2024), or even simpler parameterized functions – as well as exploring toolkits such as formal verification to better understand and assess the behavior of the learned data-driven mapping after it is trained. Another future direction includes exploring adaptive co-training of the data-driven feasibility improvement mapping with the neural network to which it will be attached (rather than training the mapping fully separately), motivated by prior results on the benefits of end-to-end learning (Cameron et al., 2022). Overall, we believe that the paradigm of data-driven feasibility mapping offers a compelling yet underexplored middle ground between penalty methods and exact feasibility enforcement, and look forward to future work that investigates alternative approaches within this space beyond simply the one presented here.

## ACKNOWLEDGMENTS

We thank Hoang Nguyen and Khai Nguyen for their input and feedback on this work. This work was supported by the U.S. National Science Foundation under award #2325956.

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

## A  HYPERPARAMETERS AND ADDITIONAL EXPERIMENTAL DETAILS

### A.1  CONSTRAINED OPTIMIZATION HYPERPARAMETERS

Hyperparameters for autoencoder training in the constrained optimization experiments (Section 4). The optimal loss weights were found using a grid search over the following values:

- $\lambda_{\text{recon}}$ options = [1.5, 2.0]
- $\lambda_{\text{feas}}$ options = [1.0, 1.5, 2.0]
- $\lambda_{\text{latent}}$ options = [1.0, 1.5]
- $\lambda_{\text{geom}}$ options = [0.025]
- $\lambda_{\text{hinge}}$ options = [0.5, 1.0]

The remaining hyperparameters for both our methods and the baselines were manually chosen via informal experimentation.

**Phase 1 Training (phase1_training.py)**

- Batch size: 256
- Epochs: 500
- Learning rate: 0.001
- Optimizer: Adam
- Training/validation split: 80% / 20%
- Hidden dimension: 64
- Number of decoders: 1
- Number of samples: 60000

**Phase 2 Training (phase2_training.py)**

- Batch size: 256
- Epochs: 150
- Learning rates: AE=0.0005, Discriminator=0.001
- Loss weights: $\lambda_{\text{recon}}$=1.0, $\lambda_{\text{feasibility}}$=1.0, $\lambda_{\text{latent}} = 1.0$, $\lambda_{\text{hinge}} = 0.1$, $\lambda_{\text{geometric}} = 0.1$
- Optimizer: Adam
- Critic steps:[4] 3
- Training/validation split: 80% / 20%
- Normalization: enabled
- Hidden dimension: 64
- Number of decoders: 1

Hyperparameters for the ablation and decoder configuration experiments described in Appendix C are in Table A.1.

### A.2  SAFETY GYM HYPERPARAMETERS

Hyperparameters for autoencoder training in the Safety Gym experiments (Section 5). Most of the hyperparameters were borrowed from the Safety Gymnasium benchmark suite implementations (Ji et al., 2023), as well as manually chosen via informal experimentation.

**Phase 1 Training (safety-gym/phase1_training.py)**

---

[4]This hyperparameter describes the number of discriminator gradient descent steps per one pass through the autoencoder.

| Shape | Exp. Type | Config | $\lambda_{recon}\star$ | $\lambda_{feas}\star$ | $\lambda_{latent}\star$ | $\lambda_{geom}\star$ | $\lambda_{hinge}\star$ | Throughput |
|---|---|---|---|---|---|---|---|---|
| blob_with_bite | coverage | Cov_10 | 1.5 | 1.0 | 1.0 | 0.025 | 0.5 | 457.44 |
| blob_with_bite | coverage | Cov_25 | 1.5 | 1.0 | 1.0 | 0.025 | 0.5 | 453.23 |
| blob_with_bite | coverage | Cov_50 | 1.5 | 1.0 | 1.0 | 0.025 | 0.5 | 446.47 |
| blob_with_bite | coverage | Cov_75 | 1.5 | 1.0 | 1.0 | 0.025 | 0.5 | 446.20 |
| blob_with_bite | capacity | W32_D2 | 1.5 | 1.0 | 1.0 | 0.025 | 0.5 | 404.14 |
| blob_with_bite | capacity | W32_D4 | 1.5 | 1.0 | 1.0 | 0.025 | 0.5 | 384.20 |
| blob_with_bite | capacity | W32_D6 | 1.5 | 1.0 | 1.0 | 0.025 | 0.5 | 368.88 |
| blob_with_bite | capacity | W64_D2 | 1.5 | 1.0 | 1.0 | 0.025 | 0.5 | 399.35 |
| blob_with_bite | capacity | W64_D4 | 1.5 | 1.0 | 1.0 | 0.025 | 0.5 | 379.72 |
| blob_with_bite | capacity | W64_D6 | 1.5 | 1.0 | 1.0 | 0.025 | 0.5 | 366.03 |
| blob_with_bite | capacity | W128_D2 | 1.5 | 1.0 | 1.0 | 0.025 | 0.5 | 394.16 |
| blob_with_bite | capacity | W128_D4 | 1.5 | 1.0 | 1.0 | 0.025 | 0.5 | 379.71 |
| blob_with_bite | capacity | W128_D6 | 1.5 | 1.0 | 1.0 | 0.025 | 0.5 | 367.23 |
| blob_with_bite | num_dec | 2_decoders | 1.5 | 1.0 | 1.0 | 0.025 | 0.5 | 349.43 |
| star_shaped | coverage | Cov_10 | 1.5 | 1.0 | 1.0 | 0.025 | 0.5 | 448.20 |
| star_shaped | coverage | Cov_25 | 1.5 | 1.0 | 1.0 | 0.025 | 0.5 | 444.26 |
| star_shaped | coverage | Cov_50 | 1.5 | 1.0 | 1.0 | 0.025 | 0.5 | 441.18 |
| star_shaped | coverage | Cov_75 | 1.5 | 1.0 | 1.0 | 0.025 | 0.5 | 436.32 |
| star_shaped | capacity | W32_D2 | 1.5 | 1.0 | 1.0 | 0.025 | 0.5 | 399.33 |
| star_shaped | capacity | W32_D4 | 1.5 | 1.0 | 1.0 | 0.025 | 0.5 | 379.74 |
| star_shaped | capacity | W32_D6 | 1.5 | 1.0 | 1.0 | 0.025 | 0.5 | 368.52 |
| star_shaped | capacity | W64_D2 | 1.5 | 1.0 | 1.0 | 0.025 | 0.5 | 394.05 |
| star_shaped | capacity | W64_D4 | 1.5 | 1.0 | 1.5 | 0.025 | 0.5 | 385.30 |
| star_shaped | capacity | W64_D6 | 1.5 | 1.0 | 1.0 | 0.025 | 0.5 | 366.63 |
| star_shaped | capacity | W128_D2 | 1.5 | 1.0 | 1.5 | 0.025 | 0.5 | 401.35 |
| star_shaped | capacity | W128_D4 | 1.5 | 1.0 | 1.0 | 0.025 | 0.5 | 382.05 |
| star_shaped | capacity | W128_D6 | 1.5 | 1.0 | 1.0 | 0.025 | 1.0 | 369.05 |
| star_shaped | num_dec | 2_decoders | 1.5 | 1.0 | 1.0 | 0.025 | 1.0 | 352.57 |
| two_moons | coverage | Cov_10 | 1.5 | 1.0 | 1.0 | 0.025 | 0.5 | 805.99 |
| two_moons | coverage | Cov_25 | 1.5 | 1.0 | 1.0 | 0.025 | 0.5 | 809.56 |
| two_moons | coverage | Cov_50 | 1.5 | 1.0 | 1.0 | 0.025 | 0.5 | 802.55 |
| two_moons | coverage | Cov_75 | 1.5 | 1.0 | 1.0 | 0.025 | 0.5 | 810.32 |
| two_moons | capacity | W32_D2 | 2.0 | 2.0 | 1.5 | 0.025 | 0.5 | 650.05 |
| two_moons | capacity | W32_D4 | 1.5 | 1.5 | 1.5 | 0.025 | 1.0 | 469.24 |
| two_moons | capacity | W32_D6 | 1.5 | 1.0 | 1.5 | 0.025 | 0.5 | 449.40 |
| two_moons | capacity | W64_D2 | 1.5 | 1.0 | 1.5 | 0.025 | 0.5 | 503.61 |
| two_moons | capacity | W64_D4 | 1.5 | 1.0 | 1.0 | 0.025 | 0.5 | 532.68 |
| two_moons | capacity | W64_D6 | 1.5 | 1.0 | 1.0 | 0.025 | 0.5 | 447.05 |
| two_moons | capacity | W128_D2 | 2.0 | 1.5 | 1.5 | 0.025 | 0.5 | 604.86 |
| two_moons | capacity | W128_D4 | 1.5 | 2.0 | 1.5 | 0.025 | 0.5 | 553.15 |
| two_moons | capacity | W128_D6 | 1.5 | 1.0 | 1.5 | 0.025 | 0.5 | 490.69 |
| two_moons | num_dec | 2_decoders | 1.5 | 1.0 | 1.5 | 0.025 | 0.5 | 433.12 |
| concentric_circles | coverage | Cov_10 | 1.5 | 1.0 | 1.0 | 0.025 | 0.5 | 834.29 |
| concentric_circles | coverage | Cov_25 | 1.5 | 1.0 | 1.0 | 0.025 | 0.5 | 829.25 |
| concentric_circles | coverage | Cov_50 | 1.5 | 1.0 | 1.0 | 0.025 | 0.5 | 736.49 |
| concentric_circles | coverage | Cov_75 | 1.5 | 1.0 | 1.0 | 0.025 | 0.5 | 830.27 |
| concentric_circles | capacity | W32_D2 | 1.5 | 1.0 | 1.0 | 0.025 | 0.5 | 862.89 |
| concentric_circles | capacity | W32_D4 | 1.5 | 1.0 | 1.0 | 0.025 | 0.5 | 703.99 |
| concentric_circles | capacity | W32_D6 | 1.5 | 1.0 | 1.0 | 0.025 | 0.5 | 637.08 |
| concentric_circles | capacity | W64_D2 | 1.5 | 1.0 | 1.5 | 0.025 | 1.0 | 764.57 |
| concentric_circles | capacity | W64_D4 | 1.5 | 1.0 | 1.0 | 0.025 | 1.0 | 791.68 |
| concentric_circles | capacity | W64_D6 | 1.5 | 1.0 | 1.0 | 0.025 | 1.0 | 708.25 |
| concentric_circles | capacity | W128_D2 | 1.5 | 1.0 | 1.0 | 0.025 | 1.0 | 895.01 |
| concentric_circles | capacity | W128_D4 | 1.5 | 1.0 | 1.0 | 0.025 | 1.0 | 788.96 |
| concentric_circles | capacity | W128_D6 | 1.5 | 1.0 | 1.0 | 0.025 | 1.0 | 737.83 |
| concentric_circles | num_dec | 2_decoders | 1.5 | 1.0 | 1.0 | 0.025 | 1.0 | 634.10 |

Table A.1: Optimal configurations for ablations and training throughput comparisons. The default decoder configuration is W64_D4.

- Batch size: 256
- Epochs: 500
- Learning rate: 0.001
- Optimizer: Adam
- Training set: 80%
- Validation set: 20%
- $\lambda_{\text{recon}}$: 1.0
- Hidden dimension: 64
- Number of decoders: 1
- Number of samples: 100000
- State dimension: 60 (safety_gym)
- Action dimension: 2 (safety_gym)
- Latent dimension: equal to action dimension

**Phase 2 Training (safety-gym/phase2_training.py)**

- Batch size: 256
- Epochs: 100
- Learning rates: AE=0.0001, Discriminator=0.0002
- Loss weights: $\lambda_{\text{recon}} = 1.0$, $\lambda_{\text{feasibility}} = 0.5$, $\lambda_{\text{latent}} = 0.5$, $\lambda_{\text{hinge}} = 0.5$, $\lambda_{\text{geometric}} = 0.1$
- Optimizer: Adam
- Validation split: 0.2
- Hidden dimension: 64
- Number of decoders: 1

## A.3 COMPUTING THE DIFFERENTIABLE SURROGATE OF CONSTRAINT VIOLATION $v(y)$

In our implementation, $v(y)$ is a Monte Carlo estimate of local infeasibility around a point $y \in \mathbb{R}^n$. Specifically, we sample $m$ Gaussian perturbations $\epsilon_i \sim \mathcal{N}(0, I)$ to generate test points $y_i = y + \sigma \epsilon_i$, and define

$$v(y) = \frac{1}{m} \sum_{i=1}^{m} \mathbf{1}\{y_i \notin \mathcal{C}\}, \tag{11}$$

where $\mathbf{1}\{\cdot\}$ is the indicator function and $\sigma > 0$ controls the scale of the local perturbations. Essentially, $v(y)$ approximates the probability that a local perturbation around $y$ results in an infeasible point.

Because the indicator function is non-differentiable, we use Gaussian smoothing to implicitly define a continuously differentiable surrogate representing the expected local infeasibility. Using zero-order gradient estimation, we can approximate the gradient of the expected surrogate by reusing the sampled perturbations. Specifically, we average the perturbation directions $\epsilon_i$ over the discrete infeasibility of their corresponding test points. This yields the gradient estimate

$$\widehat{\nabla v}(y) = -\frac{1}{m} \sum_{i=1}^{m} \mathbf{1}\{y + \sigma \epsilon_i \notin \mathcal{C}\} \frac{\epsilon_i}{\sigma}, \tag{12}$$

which provides a descent direction for gradient-based optimization.

We use this sampling-based estimator to demonstrate the efficacy of our approach for navigating topologically complex feasible regions, particularly in settings where closed-form constraint expressions or exact analytical gradients may be unavailable.

# B ADDITIONAL EXPERIMENTAL RESULTS

Tables B.2–B.4 provide experimental results for the Blob with Bite, Concentric Circles, and Star-Shaped constraint families described in Section 4, Figure 2. Results for the Two Moons constraint family are provided in the main body (Table 1).

Table B.5 provides results for the 3-, 5-, and 10-dimensional hyperball constraints $\mathcal{C} := \{x \in \mathbb{R}^d \mid 1 \le \|x\|_2^2 \le 2\}$.

| | Method | Quadratic | | | Linear | | | Dist. Min. | | |
|---|---|---|---|---|---|---|---|---|---|---|
| | | ↑ Feas (%) | ↓ Time (ms) | ↓ Gap | ↑ Feas (%) | ↓ Time (ms) | ↓ Gap | ↑ Feas (%) | ↓ Time (ms) | ↓ Gap |
| **Classical** | Projected Gradient | **100.0** | 38.73 | 0.60 | **100.0** | 19.80 | **1.04** | **100.0** | 32.34 | **0.95** |
| | | $\sigma=0.0$ | $\sigma=28.68$ | $\sigma=0.88$ | $\sigma=0.0$ | $\sigma=31.73$ | $\sigma=1.11$ | $\sigma=0.0$ | $\sigma=49.68$ | $\sigma=2.42$ |
| | Penalty Method | 59.7 | 64.76 | 0.97 | 43.7 | 38.72 | 1.29 | 55.1 | 55.01 | 5.37 |
| | | $\sigma=49.0$ | $\sigma=47.86$ | $\sigma=1.39$ | $\sigma=49.6$ | $\sigma=52.87$ | $\sigma=1.28$ | $\sigma=49.7$ | $\sigma=156.36$ | $\sigma=6.47$ |
| | Augmented Lagrangian | 58.5 | 71.64 | 0.98 | 44.5 | 42.68 | 1.33 | 56.5 | 45.49 | 5.54 |
| | | $\sigma=49.3$ | $\sigma=64.04$ | $\sigma=1.42$ | $\sigma=49.7$ | $\sigma=49.49$ | $\sigma=1.29$ | $\sigma=49.6$ | $\sigma=33.47$ | $\sigma=6.73$ |
| | Interior Point | 95.3 | 61.82 | 2.50 | 92.6 | 45.66 | 1.63 | 94.5 | 40.50 | 11.89 |
| | | $\sigma=21.2$ | $\sigma=35.51$ | $\sigma=2.61$ | $\sigma=26.2$ | $\sigma=56.51$ | $\sigma=1.66$ | $\sigma=22.7$ | $\sigma=41.82$ | $\sigma=11.35$ |
| **ML-Based** | Penalty NN | 97.1 | **0.19** | **0.50** | 90.4 | **0.14** | **1.04** | 66.0 | **0.14** | **1.24** |
| | | $\sigma=16.9$ | $\sigma=0.06$ | $\sigma=0.76$ | $\sigma=29.5$ | $\sigma=0.03$ | $\sigma=1.14$ | $\sigma=47.4$ | $\sigma=0.02$ | $\sigma=2.41$ |
| | FSNet | 99.5 | 2.66 | 1.15 | 98.7 | 2.49 | 2.09 | 99.8 | 2.31 | 13.63 |
| | | $\sigma=7.3$ | $\sigma=11.09$ | $\sigma=1.67$ | $\sigma=11.2$ | $\sigma=1.43$ | $\sigma=1.98$ | $\sigma=4.5$ | $\sigma=1.33$ | $\sigma=13.30$ |
| | Homeomorphic Projection | **100.0** | 196.631 | 4.37 | **100.0** | 196.247 | 2.58 | **100.0** | 196.021 | 13.36 |
| | | $\sigma=0.0$ | $\sigma=193.4$ | $\sigma=4.28$ | $\sigma=0.0$ | $\sigma=195.8$ | $\sigma=2.30$ | $\sigma=0.0$ | $\sigma=195.5$ | $\sigma=12.73$ |
| | FAB | **100.0** | 0.69 | 0.53 | **100.0** | 0.50 | 1.14 | **100.0** | 0.39 | 2.89 |
| | | $\sigma=0.0$ | $\sigma=1.44$ | $\sigma=0.71$ | $\sigma=0.0$ | $\sigma=0.48$ | $\sigma=0.96$ | $\sigma=0.0$ | $\sigma=0.17$ | $\sigma=4.26$ |
| | FAB: 2 Decoder Nets | 62.0 | 1.14 | 0.52 | 55.9 | 0.59 | 1.09 | 86.0 | 0.45 | 2.59 |
| | | $\sigma=48.5$ | $\sigma=4.08$ | $\sigma=0.69$ | $\sigma=49.7$ | $\sigma=0.43$ | $\sigma=0.95$ | $\sigma=34.7$ | $\sigma=0.15$ | $\sigma=4.23$ |
| | FAB: 3 Decoder Nets | 95.6 | 1.15 | 0.53 | 95.3 | 0.60 | 1.11 | 94.6 | 0.55 | 3.59 |
| | | $\sigma=20.5$ | $\sigma=3.18$ | $\sigma=0.70$ | $\sigma=21.2$ | $\sigma=0.35$ | $\sigma=0.92$ | $\sigma=22.6$ | $\sigma=0.42$ | $\sigma=4.49$ |
| | FAB: Phase 1 Only | 66.7 | 0.77 | 0.51 | 64.1 | 1.16 | 1.06 | 76.7 | 0.56 | 1.32 |
| | | $\sigma=47.1$ | $\sigma=0.34$ | $\sigma=0.62$ | $\sigma=48.0$ | $\sigma=2.88$ | $\sigma=0.83$ | $\sigma=42.3$ | $\sigma=0.13$ | $\sigma=1.48$ |
| | FAB: No Hinge Loss | **100.0** | 0.94 | 0.63 | **100.0** | 1.15 | 1.29 | **100.0** | 0.61 | 6.47 |
| | | $\sigma=0.0$ | $\sigma=1.79$ | $\sigma=0.79$ | $\sigma=0.0$ | $\sigma=1.58$ | $\sigma=1.11$ | $\sigma=0.0$ | $\sigma=0.40$ | $\sigma=5.28$ |
| | FAB: No $D_\xi$ | 80.8 | 0.74 | 0.52 | 80.6 | 0.82 | 1.11 | 77.9 | 0.73 | 4.79 |
| | | $\sigma=39.4$ | $\sigma=0.50$ | $\sigma=0.69$ | $\sigma=39.5$ | $\sigma=0.47$ | $\sigma=0.92$ | $\sigma=41.5$ | $\sigma=0.58$ | $\sigma=4.34$ |

Table B.2: Mean and std.dev. for feasibility (%), time (ms), and optimality gap for each method: Blob with Bite.

| | Method | Quadratic | | | Linear | | | Dist. Min. | | |
|---|---|---|---|---|---|---|---|---|---|---|
| | | ↑ Feas (%) | ↓ Time (ms) | ↓ Gap | ↑ Feas (%) | ↓ Time (ms) | ↓ Gap | ↑ Feas (%) | ↓ Time (ms) | ↓ Gap |
| **Classical** | Projected Gradient | 100.0 | 31.24 | 0.96 | 100.0 | 14.83 | 1.16 | 100.0 | 24.36 | 0.40 |
| | | $\sigma=0.0$ | $\sigma=13.55$ | $\sigma=1.27$ | $\sigma=0.0$ | $\sigma=14.92$ | $\sigma=1.16$ | $\sigma=0.0$ | $\sigma=17.34$ | $\sigma=0.88$ |
| | Penalty Method | 35.5 | 56.68 | 1.12 | 42.5 | 39.15 | 1.35 | 49.0 | 40.05 | 5.55 |
| | | $\sigma=47.8$ | $\sigma=44.47$ | $\sigma=1.41$ | $\sigma=49.4$ | $\sigma=63.11$ | $\sigma=1.30$ | $\sigma=50.0$ | $\sigma=31.26$ | $\sigma=6.60$ |
| | Augmented Lagrangian | 37.2 | 57.59 | 1.16 | 42.1 | 52.09 | 1.39 | 49.3 | 43.93 | 5.70 |
| | | $\sigma=48.3$ | $\sigma=43.84$ | $\sigma=1.43$ | $\sigma=49.4$ | $\sigma=85.00$ | $\sigma=1.30$ | $\sigma=50.0$ | $\sigma=46.15$ | $\sigma=6.91$ |
| | Interior Point | 93.4 | 54.53 | 2.67 | 89.9 | 39.76 | 1.78 | 92.6 | 44.17 | 11.89 |
| | | $\sigma=24.8$ | $\sigma=34.56$ | $\sigma=2.71$ | $\sigma=30.2$ | $\sigma=29.55$ | $\sigma=1.79$ | $\sigma=26.2$ | $\sigma=38.55$ | $\sigma=11.28$ |
| **ML-Based** | Penalty NN | 92.3 | 0.24 | 0.92 | 96.2 | 0.14 | 1.06 | 51.6 | 0.14 | 0.97 |
| | | $\sigma=26.7$ | $\sigma=0.51$ | $\sigma=1.19$ | $\sigma=19.1$ | $\sigma=0.03$ | $\sigma=1.17$ | $\sigma=50.0$ | $\sigma=0.02$ | $\sigma=2.08$ |
| | FSNet | 99.9 | 1.75 | 1.29 | 98.5 | 3.04 | 1.72 | 99.1 | 2.03 | 9.72 |
| | | $\sigma=2.6$ | $\sigma=1.01$ | $\sigma=1.72$ | $\sigma=12.0$ | $\sigma=6.11$ | $\sigma=2.01$ | $\sigma=9.3$ | $\sigma=0.97$ | $\sigma=14.52$ |
| | Homeomorphic Projection | 93.7 | 166 | 4.83 | 93.4 | 168 | 2.56 | 93.4 | 173 | 14.08 |
| | | $\sigma=24.4$ | $\sigma=86$ | $\sigma=4.31$ | $\sigma=24.8$ | $\sigma=87$ | $\sigma=2.32$ | $\sigma=24.8$ | $\sigma=90$ | $\sigma=13.51$ |
| | FAB | 100.0 | 0.45 | 1.75 | 100.0 | 0.38 | 1.48 | 99.9 | 0.48 | 5.72 |
| | | $\sigma=0.0$ | $\sigma=0.21$ | $\sigma=1.67$ | $\sigma=0.0$ | $\sigma=0.13$ | $\sigma=1.36$ | $\sigma=2.6$ | $\sigma=0.36$ | $\sigma=7.72$ |
| | FAB: 2 Decoder Nets | 99.8 | 0.50 | 1.49 | 92.7 | 0.49 | 1.52 | 81.4 | 0.52 | 1.54 |
| | | $\sigma=4.5$ | $\sigma=0.17$ | $\sigma=1.44$ | $\sigma=27.5$ | $\sigma=0.30$ | $\sigma=1.33$ | $\sigma=38.9$ | $\sigma=0.30$ | $\sigma=1.53$ |
| | FAB: 3 Decoder Nets | 99.3 | 0.86 | 1.50 | 96.9 | 0.61 | 1.60 | 82.5 | 0.71 | 1.25 |
| | | $\sigma=8.1$ | $\sigma=1.94$ | $\sigma=1.43$ | $\sigma=17.4$ | $\sigma=0.44$ | $\sigma=1.47$ | $\sigma=38.0$ | $\sigma=1.21$ | $\sigma=1.25$ |
| | FAB: Phase 1 Only | 99.1 | 0.77 | 0.89 | 98.0 | 0.54 | 1.12 | 25.9 | 0.51 | 4.61 |
| | | $\sigma=3.6$ | $\sigma=2.51$ | $\sigma=0.82$ | $\sigma=14.0$ | $\sigma=0.11$ | $\sigma=0.81$ | $\sigma=43.8$ | $\sigma=0.04$ | $\sigma=3.63$ |
| | FAB: No Hinge Loss | 99.1 | 0.74 | 1.30 | 98.1 | 0.58 | 1.48 | 68.7 | 0.51 | 7.08 |
| | | $\sigma=9.9$ | $\sigma=0.57$ | $\sigma=1.29$ | $\sigma=13.5$ | $\sigma=0.21$ | $\sigma=1.29$ | $\sigma=46.4$ | $\sigma=0.03$ | $\sigma=7.85$ |
| | FAB: No $D_\xi$ | 98.3 | 0.74 | 1.02 | 98.5 | 0.72 | 1.23 | 82.7 | 0.51 | 1.58 |
| | | $\sigma=12.8$ | $\sigma=0.67$ | $\sigma=0.92$ | $\sigma=12.3$ | $\sigma=2.67$ | $\sigma=0.96$ | $\sigma=37.9$ | $\sigma=0.03$ | $\sigma=2.15$ |

Table B.3: Mean and std. dev. for feasibility (%), time (ms), and optimality gap for each method: Concentric Circles constraint family. 5 seeds, 300 problems each per problem per objective type.

| | Method | Quadratic | | | Linear | | | Dist. Min. | | |
|---|---|---|---|---|---|---|---|---|---|---|
| | | ↑ Feas (%) | ↓ Time (ms) | ↓ Gap | ↑ Feas (%) | ↓ Time (ms) | ↓ Gap | ↑ Feas (%) | ↓ Time (ms) | ↓ Gap |
| **Classical** | Projected Gradient | 100.0 | 38.41 | 0.34 | 100.0 | 18.99 | 0.78 | 100.0 | 31.18 | 1.01 |
| | | $\sigma=0.0$ | $\sigma=39.50$ | $\sigma=0.41$ | $\sigma=0.0$ | $\sigma=10.97$ | $\sigma=0.69$ | $\sigma=0.0$ | $\sigma=27.47$ | $\sigma=1.86$ |
| | Penalty Method | 75.8 | 61.37 | 0.87 | 38.0 | 40.03 | 1.17 | 57.3 | 39.41 | 4.84 |
| | | $\sigma=42.8$ | $\sigma=33.75$ | $\sigma=1.32$ | $\sigma=48.5$ | $\sigma=33.05$ | $\sigma=1.15$ | $\sigma=49.5$ | $\sigma=13.16$ | $\sigma=5.99$ |
| | Augmented Lagrangian | 75.0 | 70.70 | 0.89 | 40.4 | 39.49 | 1.19 | 58.1 | 48.25 | 4.99 |
| | | $\sigma=43.3$ | $\sigma=74.02$ | $\sigma=1.38$ | $\sigma=49.1$ | $\sigma=20.97$ | $\sigma=1.10$ | $\sigma=49.3$ | $\sigma=32.70$ | $\sigma=6.17$ |
| | Interior Point | 96.6 | 70.65 | 1.77 | 92.9 | 40.25 | 1.45 | 95.0 | 53.22 | 9.43 |
| | | $\sigma=18.1$ | $\sigma=75.77$ | $\sigma=1.99$ | $\sigma=25.7$ | $\sigma=21.66$ | $\sigma=1.46$ | $\sigma=21.8$ | $\sigma=62.60$ | $\sigma=8.70$ |
| **ML-Based** | Penalty NN | 98.2 | 0.20 | 0.27 | 83.5 | 0.14 | 0.80 | 63.9 | 0.14 | 1.40 |
| | | $\sigma=13.3$ | $\sigma=0.07$ | $\sigma=0.32$ | $\sigma=37.1$ | $\sigma=0.03$ | $\sigma=0.71$ | $\sigma=48.0$ | $\sigma=0.02$ | $\sigma=2.71$ |
| | FSNet | 99.9 | 3.13 | 1.03 | 99.7 | 3.76 | 1.05 | 99.7 | 5.41 | 7.24 |
| | | $\sigma=3.6$ | $\sigma=3.00$ | $\sigma=1.91$ | $\sigma=5.2$ | $\sigma=2.06$ | $\sigma=1.04$ | $\sigma=5.2$ | $\sigma=11.90$ | $\sigma=7.61$ |
| | Homeomorphic Projection | 100.0 | 128.653 | 3.70 | 100.0 | 126.350 | 2.35 | 100.0 | 126.455 | 12.49 |
| | | $\sigma=0.0$ | $\sigma=48.4$ | $\sigma=3.35$ | $\sigma=0.0$ | $\sigma=48.9$ | $\sigma=2.07$ | $\sigma=0.0$ | $\sigma=48.9$ | $\sigma=11.84$ |
| | FAB | 99.9 | 0.56 | 0.34 | 99.9 | 0.60 | 0.77 | 95.4 | 0.64 | 2.01 |
| | | $\sigma=2.6$ | $\sigma=0.45$ | $\sigma=0.40$ | $\sigma=2.6$ | $\sigma=1.61$ | $\sigma=0.65$ | $\sigma=20.9$ | $\sigma=0.93$ | $\sigma=1.96$ |
| | FAB: 2 Decoder | 100.0 | 0.60 | 0.37 | 99.5 | 0.49 | 0.73 | 94.7 | 0.71 | 2.30 |
| | | $\sigma=0.0$ | $\sigma=0.32$ | $\sigma=0.43$ | $\sigma=7.3$ | $\sigma=0.26$ | $\sigma=0.59$ | $\sigma=22.3$ | $\sigma=0.83$ | $\sigma=2.28$ |
| | FAB: 3 Decoder | 99.9 | 0.65 | 0.34 | 99.9 | 0.55 | 0.75 | 94.5 | 0.93 | 2.02 |
| | | $\sigma=2.6$ | $\sigma=0.29$ | $\sigma=0.39$ | $\sigma=3.6$ | $\sigma=0.24$ | $\sigma=0.62$ | $\sigma=22.9$ | $\sigma=2.39$ | $\sigma=2.00$ |
| | FAB: Phase 1 Only | 100.0 | 0.61 | 0.34 | 100.0 | 0.54 | 0.76 | 100.0 | 0.53 | 3.05 |
| | | $\sigma=0.0$ | $\sigma=0.08$ | $\sigma=0.41$ | $\sigma=0.0$ | $\sigma=0.08$ | $\sigma=0.65$ | $\sigma=0.0$ | $\sigma=0.08$ | $\sigma=3.04$ |
| | FAB: No Hinge Loss | 100.0 | 0.60 | 0.38 | 100.0 | 0.54 | 0.83 | 100.0 | 0.52 | 5.00 |
| | | $\sigma=0.0$ | $\sigma=0.07$ | $\sigma=0.49$ | $\sigma=0.0$ | $\sigma=0.07$ | $\sigma=0.75$ | $\sigma=0.0$ | $\sigma=0.04$ | $\sigma=4.27$ |
| | FAB: No $D_\xi$ | 100.0 | 0.86 | 0.35 | 100.0 | 0.54 | 0.78 | 96.8 | 0.52 | 1.62 |
| | | $\sigma=0.0$ | $\sigma=2.05$ | $\sigma=0.40$ | $\sigma=0.0$ | $\sigma=0.07$ | $\sigma=0.65$ | $\sigma=17.6$ | $\sigma=0.04$ | $\sigma=1.64$ |

Table B.4: Mean and std. dev. for feasibility (%), time (ms), and optimality gap for each method: Star Shaped constraint family. 5 seeds, 300 problems each per problem per objective type.

| Constraint Dimension | Quadratic | | | Linear | | | Dist. Min. | | |
|---|---|---|---|---|---|---|---|---|---|
| | ↑ Feas (%) | ↓ Time (ms) | ↓ Gap | ↑ Feas (%) | ↓ Time (ms) | ↓ Gap | ↑ Feas (%) | ↓ Time (ms) | ↓ Gap |
| Penalty NN: 3D | 86.6 | 0.12 | 1.6316 | 77.8 | 0.08 | 1.5220 | 55.0 | 0.09 | 6.8719 |
| | $\sigma= 34.1$ | $\sigma= 0.06$ | $\sigma= 1.6897$ | $\sigma= 41.6$ | $\sigma= 0.02$ | $\sigma= 1.4711$ | $\sigma= 49.7$ | $\sigma= 0.03$ | $\sigma= 9.6887$ |
| Penalty NN: 5D | 88.3 | 0.12 | 1.8296 | 85.8 | 0.08 | 1.7951 | 28.4 | 0.08 | 12.5372 |
| | $\sigma= 32.1$ | $\sigma= 0.03$ | $\sigma= 1.7225$ | $\sigma= 34.9$ | $\sigma= 0.02$ | $\sigma= 1.1600$ | $\sigma= 45.1$ | $\sigma= 0.02$ | $\sigma= 12.5638$ |
| Penalty NN: 10D | 64.3 | 0.13 | 3.1529 | 80.4 | 0.09 | 3.6322 | 0.5 | 0.08 | 49.9375 |
| | $\sigma= 47.9$ | $\sigma= 0.09$ | $\sigma= 2.2872$ | $\sigma= 39.7$ | $\sigma= 0.05$ | $\sigma= 1.6177$ | $\sigma= 7.3$ | $\sigma= 0.01$ | $\sigma= 27.8050$ |
| FAB: 3D | 100.00 | 0.81 | 2.08 | 99.47 | 0.77 | 1.66 | 97.67 | 0.80 | 6.28 |
| | $\sigma= 0.00$ | $\sigma= 0.01$ | $\sigma= 1.74$ | $\sigma= 0.07$ | $\sigma= 0.04$ | $\sigma= 1.40$ | $\sigma= 0.15$ | $\sigma= 0.04$ | $\sigma= 6.98$ |
| FAB: 5D | 90.50 | 0.83 | 2.08 | 95.20 | 0.74 | 1.94 | 96.47 | 0.77 | 4.30 |
| | $\sigma= 0.32$ | $\sigma= 0.03$ | $\sigma= 1.52$ | $\sigma= 0.22$ | $\sigma= 0.02$ | $\sigma= 1.20$ | $\sigma= 0.18$ | $\sigma= 0.04$ | $\sigma= 4.77$ |
| FAB: 10D | 100.00 | 0.81 | 10.97 | 100.00 | 0.74 | 1.94 | 100.00 | 0.74 | 12.69 |
| | $\sigma= 0.00$ | $\sigma= 0.01$ | $\sigma= 6.45$ | $\sigma= 0.00$ | $\sigma= 0.01$ | $\sigma= 1.48$ | $\sigma= 0.00$ | $\sigma= 0.01$ | $\sigma= 8.96$ |

Table B.5: Mean and std. dev. for feasibility (%), time (ms), and optimality gap for Penalty NN and FAB run on the 3-, 5-, and 10-dimensional hyperball constraint sets.

## C  ADDITIONAL ABLATIONS AND CONFIGURATIONS

In Tables C.6–C.9, we provide results for two additional types of variants of our FAB method for the 2-dimensional constrained optimization settings:

- **Incomplete coverage** ablations test the performance of our method when the training data only consists of part of the constraint set (10%, 25%, 50%, and 75%).

- **Decoder capacity** experiments test the performance of multiple decoder depth and width configurations. W indicates decoder width, while D indicates decoder depth.

For the incomplete coverage ablations, we find that FAB is generally able to maintain feasibility even when the feasible set is not fully covered during training, achieving relatively high feasibility rates across all ablations (with the lowest being $80.1 \pm 0.0\%$ for Concentric Circles Cov-75 and $76.5 \pm 24.7\%$ for Two Moons Cov-25, and with other scores all above 90% and largely at 100%). Notably, the 10% coverage ablation achieves 100% feasibility across all feasibility sets, likely because the feasibility sets may become topologically simpler if the coverage is low enough, which means they are easier for the autoencoder to learn. Optimality gaps vary across the experiments, depending on whether the optimal solution happens to lie within the part of the constraint set that is sampled. (The reported optimality gaps also vary based on whether feasibility is maintained, as infeasible solutions can achieve artificially low "optimality gaps.") Overall, we find that while perfect coverage of the constraint set is not necessary, better coverage of the constraint set during training unsurprisingly enables better representation of the full constraint set and thereby better end-to-end learning outcomes in general.

For the decoder capacity experiments, we find that performance varies depending on the problem type. Therefore, while we did not systematically tune the decoder width and depth during our main experiments, tuning these parameters may indeed be beneficial in general for overall performance.

| Method | | Quadratic | | | Linear | | | Dist. Min. | | |
|---|---|---|---|---|---|---|---|---|---|---|
| | | ↑ Feas (%) | ↓ Time (ms) | ↓ Gap | ↑ Feas (%) | ↓ Time (ms) | ↓ Gap | ↑ Feas (%) | ↓ Time (ms) | ↓ Gap |
| **Incomplete Coverage** | Cov-10 | 100.0 | 1.12 | 0.82 | 100.0 | 0.87 | 1.53 | 100.0 | 0.85 | 10.25 |
| | | $\sigma= 0.0$ | $\sigma= 1.10$ | $\sigma= 1.02$ | $\sigma= 0.0$ | $\sigma= 0.04$ | $\sigma= 1.39$ | $\sigma= 0.0$ | $\sigma= 0.04$ | $\sigma= 7.33$ |
| | Cov-25 | 100.0 | 1.08 | 0.81 | 100.0 | 0.86 | 1.52 | 100.0 | 0.85 | 10.15 |
| | | $\sigma= 0.0$ | $\sigma= 0.09$ | $\sigma= 1.01$ | $\sigma= 0.0$ | $\sigma= 0.01$ | $\sigma= 1.39$ | $\sigma= 0.0$ | $\sigma= 0.03$ | $\sigma= 7.20$ |
| | Cov-50 | 100.0 | 1.08 | 0.80 | 100.0 | 0.86 | 1.47 | 100.0 | 0.85 | 9.15 |
| | | $\sigma= 0.0$ | $\sigma= 0.09$ | $\sigma= 0.98$ | $\sigma= 0.0$ | $\sigma= 0.01$ | $\sigma= 1.30$ | $\sigma= 0.0$ | $\sigma= 0.03$ | $\sigma= 6.97$ |
| | Cov-75 | 100.0 | 1.08 | 0.81 | 100.0 | 0.86 | 1.52 | 100.0 | 0.85 | 9.46 |
| | | $\sigma= 0.0$ | $\sigma= 0.09$ | $\sigma= 1.01$ | $\sigma= 0.0$ | $\sigma= 0.03$ | $\sigma= 1.38$ | $\sigma= 0.0$ | $\sigma= 0.03$ | $\sigma= 6.86$ |
| **Decoder capacity** | W32-D2 | 99.5 | 1.09 | 0.62 | 98.5 | 0.79 | 1.21 | 72.5 | 0.80 | 6.36 |
| | | $\sigma= 6.8$ | $\sigma= 1.48$ | $\sigma= 0.80$ | $\sigma= 12.0$ | $\sigma= 0.03$ | $\sigma= 1.03$ | $\sigma= 44.6$ | $\sigma= 0.03$ | $\sigma= 5.14$ |
| | W32-D4 | 98.6 | 1.11 | 0.67 | 98.5 | 0.86 | 1.35 | 77.3 | 0.88 | 9.52 |
| | | $\sigma= 11.8$ | $\sigma= 0.08$ | $\sigma= 0.88$ | $\sigma= 12.3$ | $\sigma= 0.03$ | $\sigma= 1.23$ | $\sigma= 41.9$ | $\sigma= 0.03$ | $\sigma= 7.37$ |
| | W32-D6 | 99.4 | 1.17 | 0.72 | 98.7 | 0.93 | 1.20 | 96.8 | 0.95 | 6.54 |
| | | $\sigma= 7.7$ | $\sigma= 0.08$ | $\sigma= 0.81$ | $\sigma= 11.5$ | $\sigma= 0.03$ | $\sigma= 1.06$ | $\sigma= 17.6$ | $\sigma= 0.03$ | $\sigma= 5.17$ |
| | W64-D2 | 100.0 | 1.03 | 0.58 | 100.0 | 0.79 | 1.17 | 100.0 | 0.80 | 6.19 |
| | | $\sigma= 0.0$ | $\sigma= 0.07$ | $\sigma= 0.77$ | $\sigma= 0.0$ | $\sigma= 0.03$ | $\sigma= 0.99$ | $\sigma= 0.0$ | $\sigma= 0.02$ | $\sigma= 5.09$ |
| | W64-D6 | 67.3 | 1.16 | 0.76 | 68.6 | 0.93 | 1.42 | 52.2 | 0.94 | 7.10 |
| | | $\sigma= 46.9$ | $\sigma= 0.08$ | $\sigma= 0.93$ | $\sigma= 46.4$ | $\sigma= 0.03$ | $\sigma= 1.28$ | $\sigma= 50.0$ | $\sigma= 0.02$ | $\sigma= 5.50$ |
| | W128-D2 | 99.9 | 1.04 | 0.57 | 99.7 | 0.79 | 1.22 | 95.3 | 0.80 | 6.13 |
| | | $\sigma= 3.7$ | $\sigma= 0.07$ | $\sigma= 0.76$ | $\sigma= 5.2$ | $\sigma= 0.03$ | $\sigma= 1.03$ | $\sigma= 21.2$ | $\sigma= 0.02$ | $\sigma= 5.08$ |
| | W128-D4 | 100.0 | 1.10 | 0.82 | 100.0 | 0.86 | 1.51 | 100.0 | 0.87 | 9.70 |
| | | $\sigma= 0.0$ | $\sigma= 0.07$ | $\sigma= 0.98$ | $\sigma= 0.0$ | $\sigma= 0.03$ | $\sigma= 1.36$ | $\sigma= 0.0$ | $\sigma= 0.02$ | $\sigma= 7.01$ |
| | W128-D6 | 100.0 | 1.16 | 0.82 | 100.0 | 0.93 | 1.55 | 100.0 | 0.94 | 10.26 |
| | | $\sigma= 0.0$ | $\sigma= 0.08$ | $\sigma= 1.02$ | $\sigma= 0.0$ | $\sigma= 0.03$ | $\sigma= 1.41$ | $\sigma= 0.0$ | $\sigma= 0.06$ | $\sigma= 7.38$ |

Table C.6: Mean and std. dev. for feasibility (%), time (ms), and optimality gap for each ablation: Blob with Bite.

| Method | Quadratic | | | Linear | | | Dist. Min. | | |
|---|---|---|---|---|---|---|---|---|---|
| | ↑ Feas (%) | ↓ Time (ms) | ↓ Gap | ↑ Feas (%) | ↓ Time (ms) | ↓ Gap | ↑ Feas (%) | ↓ Time (ms) | ↓ Gap |
| **Incomplete Coverage** Cov-10 | 100.0 $\sigma = 0.0$ | 1.37 $\sigma = 0.33$ | 1.62 $\sigma = 1.54$ | 100.0 $\sigma = 0.0$ | 0.52 $\sigma = 0.04$ | 1.70 $\sigma = 1.61$ | 100.0 $\sigma = 0.0$ | 0.53 $\sigma = 0.03$ | 11.54 $\sigma = 9.31$ |
| Cov-25 | 100.0 $\sigma = 0.0$ | 1.36 $\sigma = 0.12$ | 2.75 $\sigma = 2.42$ | 100.0 $\sigma = 0.0$ | 0.52 $\sigma = 0.02$ | 1.69 $\sigma = 1.60$ | 100.0 $\sigma = 0.0$ | 0.54 $\sigma = 0.47$ | 10.71 $\sigma = 9.97$ |
| Cov-50 | 100.0 $\sigma = 0.0$ | 1.36 $\sigma = 0.13$ | 2.43 $\sigma = 2.16$ | 100.0 $\sigma = 0.0$ | 0.52 $\sigma = 0.02$ | 1.60 $\sigma = 1.51$ | 100.0 $\sigma = 0.0$ | 0.53 $\sigma = 0.02$ | 9.69 $\sigma = 11.02$ |
| Cov-75 | 80.1 $\sigma = 0.0$ | 1.34 $\sigma = 0.14$ | 1.39 $\sigma = 1.38$ | 84.3 $\sigma = 0.0$ | 0.52 $\sigma = 0.02$ | 1.16 $\sigma = 1.02$ | 35.5 $\sigma = 0.0$ | 0.53 $\sigma = 0.02$ | 4.29 $\sigma = 5.27$ |
| **Decoder capacity** W32-D2 | 99.5 $\sigma = 0.0$ | 0.94 $\sigma = 0.37$ | 0.95 $\sigma = 0.80$ | 94.5 $\sigma = 0.0$ | 0.48 $\sigma = 0.02$ | 1.18 $\sigma = 0.89$ | 14.7 $\sigma = 0.0$ | 0.47 $\sigma = 0.02$ | 1.82 $\sigma = 1.65$ |
| W32-D4 | 99.0 $\sigma = 0.0$ | 0.97 $\sigma = 0.19$ | 1.36 $\sigma = 1.31$ | 95.8 $\sigma = 0.0$ | 0.52 $\sigma = 0.01$ | 1.32 $\sigma = 1.11$ | 23.7 $\sigma = 0.0$ | 0.51 $\sigma = 0.01$ | 3.75 $\sigma = 6.07$ |
| W32-D6 | 99.1 $\sigma = 0.0$ | 1.02 $\sigma = 0.19$ | 1.33 $\sigma = 1.27$ | 91.0 $\sigma = 0.0$ | 0.55 $\sigma = 0.01$ | 1.27 $\sigma = 1.08$ | 23.0 $\sigma = 0.0$ | 0.55 $\sigma = 0.01$ | 3.17 $\sigma = 5.37$ |
| W64-D2 | 98.7 $\sigma = 0.0$ | 0.94 $\sigma = 0.19$ | 0.96 $\sigma = 0.83$ | 95.3 $\sigma = 0.0$ | 0.47 $\sigma = 0.01$ | 1.19 $\sigma = 0.86$ | 14.7 $\sigma = 0.0$ | 0.47 $\sigma = 0.01$ | 1.44 $\sigma = 1.00$ |
| W64-D4 | 98.7 $\sigma = 0.0$ | 0.98 $\sigma = 0.19$ | 1.25 $\sigma = 1.16$ | 98.1 $\sigma = 0.0$ | 0.51 $\sigma = 0.01$ | 1.38 $\sigma = 1.23$ | 24.5 $\sigma = 0.0$ | 0.51 $\sigma = 0.01$ | 3.08 $\sigma = 4.28$ |
| W64-D6 | 98.9 $\sigma = 0.0$ | 1.01 $\sigma = 0.18$ | 1.38 $\sigma = 1.30$ | 94.1 $\sigma = 0.0$ | 0.55 $\sigma = 0.01$ | 1.30 $\sigma = 1.10$ | 21.1 $\sigma = 0.0$ | 0.55 $\sigma = 0.01$ | 2.89 $\sigma = 3.90$ |
| W128-D2 | 99.9 $\sigma = 0.0$ | 0.93 $\sigma = 0.19$ | 1.40 $\sigma = 1.32$ | 98.3 $\sigma = 0.0$ | 0.48 $\sigma = 0.01$ | 1.43 $\sigma = 1.25$ | 27.3 $\sigma = 0.0$ | 0.48 $\sigma = 0.01$ | 3.91 $\sigma = 5.96$ |
| W128-D4 | 98.8 $\sigma = 0.0$ | 0.99 $\sigma = 0.20$ | 1.32 $\sigma = 1.22$ | 94.1 $\sigma = 0.0$ | 0.52 $\sigma = 0.01$ | 1.36 $\sigma = 1.18$ | 21.7 $\sigma = 0.0$ | 0.52 $\sigma = 0.01$ | 2.95 $\sigma = 3.96$ |
| W128-D6 | 99.5 $\sigma = 0.0$ | 1.03 $\sigma = 0.21$ | 1.35 $\sigma = 1.30$ | 96.9 $\sigma = 0.0$ | 0.56 $\sigma = 0.01$ | 1.40 $\sigma = 1.22$ | 16.4 $\sigma = 0.0$ | 0.56 $\sigma = 0.01$ | 3.22 $\sigma = 5.07$ |

Table C.7: Mean and std. dev. for feasibility (%), time (ms), and optimality gap for each ablation: Concentric Circles.

| Method | | Quadratic | | | Linear | | | Dist. Min. | | |
|---|---|---|---|---|---|---|---|---|---|---|
| | | ↑ Feas (%) | ↓ Time (ms) | ↓ Gap | ↑ Feas (%) | ↓ Time (ms) | ↓ Gap | ↑ Feas (%) | ↓ Time (ms) | ↓ Gap |
| **Incomplete Coverage** | Cov-10 | 100.0 | 1.06 | 1.30 | 100.0 | 0.87 | 1.43 | 100.0 | 0.88 | 9.57 |
| | | $\sigma = 0.0$ | $\sigma = 0.11$ | $\sigma = 1.25$ | $\sigma = 0.0$ | $\sigma = 0.02$ | $\sigma = 1.40$ | $\sigma = 0.0$ | $\sigma = 0.03$ | $\sigma = 7.80$ |
| | Cov-25 | 100.0 | 1.04 | 1.21 | 100.0 | 0.86 | 1.00 | 100.0 | 0.87 | 5.31 |
| | | $\sigma = 0.0$ | $\sigma = 0.09$ | $\sigma = 1.36$ | $\sigma = 0.0$ | $\sigma = 0.01$ | $\sigma = 0.99$ | $\sigma = 0.0$ | $\sigma = 0.02$ | $\sigma = 4.25$ |
| | Cov-50 | 100.0 | 1.05 | 0.84 | 100.0 | 0.86 | 0.92 | 100.0 | 0.87 | 5.93 |
| | | $\sigma = 0.0$ | $\sigma = 0.09$ | $\sigma = 0.93$ | $\sigma = 0.0$ | $\sigma = 0.01$ | $\sigma = 0.91$ | $\sigma = 0.0$ | $\sigma = 0.03$ | $\sigma = 5.43$ |
| | Cov-75 | 100.0 | 1.05 | 0.75 | 100.0 | 0.86 | 0.82 | 100.0 | 0.87 | 4.72 |
| | | $\sigma = 0.0$ | $\sigma = 0.09$ | $\sigma = 0.78$ | $\sigma = 0.0$ | $\sigma = 0.01$ | $\sigma = 0.75$ | $\sigma = 0.0$ | $\sigma = 0.03$ | $\sigma = 4.39$ |
| **Decoder capacity** | W32-D2 | 100.0 | 1.01 | 0.41 | 100.0 | 0.81 | 0.76 | 100.0 | 0.81 | 3.66 |
| | | $\sigma = 0.0$ | $\sigma = 0.11$ | $\sigma = 0.48$ | $\sigma = 0.0$ | $\sigma = 0.03$ | $\sigma = 0.66$ | $\sigma = 0.0$ | $\sigma = 0.03$ | $\sigma = 3.48$ |
| | W32-D4 | 100.0 | 1.07 | 0.37 | 100.0 | 0.88 | 0.79 | 100.0 | 0.88 | 3.55 |
| | | $\sigma = 0.0$ | $\sigma = 0.09$ | $\sigma = 0.44$ | $\sigma = 0.0$ | $\sigma = 0.05$ | $\sigma = 0.68$ | $\sigma = 0.0$ | $\sigma = 0.04$ | $\sigma = 3.46$ |
| | W32-D6 | 100.0 | 1.13 | 0.39 | 100.0 | 0.95 | 0.77 | 100.0 | 0.95 | 3.77 |
| | | $\sigma = 0.0$ | $\sigma = 0.09$ | $\sigma = 0.46$ | $\sigma = 0.0$ | $\sigma = 0.03$ | $\sigma = 0.66$ | $\sigma = 0.0$ | $\sigma = 0.02$ | $\sigma = 3.58$ |
| | W64-D2 | 100.0 | 1.00 | 0.39 | 100.0 | 0.80 | 0.76 | 100.0 | 0.81 | 3.74 |
| | | $\sigma = 0.0$ | $\sigma = 0.09$ | $\sigma = 0.46$ | $\sigma = 0.0$ | $\sigma = 0.02$ | $\sigma = 0.65$ | $\sigma = 0.0$ | $\sigma = 0.02$ | $\sigma = 3.54$ |
| | W64-D6 | 100.0 | 1.14 | 0.87 | 100.0 | 0.95 | 1.06 | 100.0 | 0.95 | 7.90 |
| | | $\sigma = 0.0$ | $\sigma = 0.09$ | $\sigma = 1.10$ | $\sigma = 0.0$ | $\sigma = 0.03$ | $\sigma = 1.11$ | $\sigma = 0.0$ | $\sigma = 0.02$ | $\sigma = 7.51$ |
| | W128-D2 | 100.0 | 0.99 | 0.38 | 100.0 | 0.81 | 0.77 | 100.0 | 0.81 | 3.62 |
| | | $\sigma = 0.0$ | $\sigma = 0.09$ | $\sigma = 0.44$ | $\sigma = 0.0$ | $\sigma = 0.02$ | $\sigma = 0.67$ | $\sigma = 0.0$ | $\sigma = 0.02$ | $\sigma = 3.48$ |
| | W128-D4 | 100.0 | 1.05 | 0.38 | 100.0 | 0.88 | 0.83 | 100.0 | 0.88 | 3.51 |
| | | $\sigma = 0.0$ | $\sigma = 0.09$ | $\sigma = 0.44$ | $\sigma = 0.0$ | $\sigma = 0.01$ | $\sigma = 0.73$ | $\sigma = 0.0$ | $\sigma = 0.02$ | $\sigma = 3.34$ |
| | W128-D6 | 100.0 | 1.14 | 0.37 | 100.0 | 0.96 | 0.74 | 100.0 | 0.95 | 3.40 |
| | | $\sigma = 0.0$ | $\sigma = 0.09$ | $\sigma = 0.44$ | $\sigma = 0.0$ | $\sigma = 0.01$ | $\sigma = 0.63$ | $\sigma = 0.0$ | $\sigma = 0.02$ | $\sigma = 3.39$ |

Table C.8: Mean and std. dev. for feasibility (%), time (ms), and optimality gap for each ablation: Star Shaped.

| Method | | Quadratic | | | Linear | | | Dist. Min. | | |
|---|---|---|---|---|---|---|---|---|---|---|
| | | ↑ Feas (%) | ↓ Time (ms) | ↓ Gap | ↑ Feas (%) | ↓ Time (ms) | ↓ Gap | ↑ Feas (%) | ↓ Time (ms) | ↓ Gap |
| **Incomplete Coverage** | Cov-10 | 100.0 | 1.48 | 1.64 | 100.0 | 0.52 | 1.70 | 100.0 | 0.53 | 11.54 |
| | | $\sigma=0.0$ | $\sigma=1.05$ | $\sigma=1.50$ | $\sigma=0.0$ | $\sigma=0.04$ | $\sigma=1.61$ | $\sigma=0.0$ | $\sigma=0.03$ | $\sigma=9.31$ |
| | Cov-25 | 76.5 | 1.46 | 2.15 | 81.1 | 0.52 | 1.69 | 50.5 | 0.54 | 10.71 |
| | | $\sigma=24.7$ | $\sigma=0.15$ | $\sigma=1.92$ | $\sigma=0.0$ | $\sigma=0.02$ | $\sigma=1.60$ | $\sigma=0.0$ | $\sigma=0.47$ | $\sigma=9.97$ |
| | Cov-50 | 92.7 | 1.45 | 0.82 | 100.0 | 0.52 | 1.60 | 64.4 | 0.53 | 9.69 |
| | | $\sigma=26.5$ | $\sigma=0.11$ | $\sigma=0.90$ | $\sigma=0.0$ | $\sigma=0.02$ | $\sigma=1.51$ | $\sigma=0.0$ | $\sigma=0.02$ | $\sigma=11.02$ |
| | Cov-75 | 91.2 | 1.46 | 0.90 | 84.3 | 0.52 | 1.16 | 23.1 | 0.53 | 4.29 |
| | | $\sigma=28.3$ | $\sigma=0.13$ | $\sigma=1.02$ | $\sigma=0.0$ | $\sigma=0.02$ | $\sigma=1.02$ | $\sigma=0.0$ | $\sigma=0.02$ | $\sigma=5.27$ |
| **Decoder capacity** | W32-D2 | 99.5 | 0.94 | 0.95 | 94.5 | 0.48 | 1.18 | 14.7 | 0.47 | 1.82 |
| | | $\sigma=0.0$ | $\sigma=0.37$ | $\sigma=0.80$ | $\sigma=0.0$ | $\sigma=0.02$ | $\sigma=0.89$ | $\sigma=0.0$ | $\sigma=0.02$ | $\sigma=1.65$ |
| | W32-D4 | 99.0 | 0.97 | 1.36 | 95.8 | 0.52 | 1.32 | 23.7 | 0.51 | 3.75 |
| | | $\sigma=0.0$ | $\sigma=0.19$ | $\sigma=1.31$ | $\sigma=0.0$ | $\sigma=0.01$ | $\sigma=1.11$ | $\sigma=0.0$ | $\sigma=0.01$ | $\sigma=6.07$ |
| | W32-D6 | 99.1 | 1.02 | 1.33 | 91.0 | 0.55 | 1.27 | 23.0 | 0.55 | 3.17 |
| | | $\sigma=0.0$ | $\sigma=0.19$ | $\sigma=1.27$ | $\sigma=0.0$ | $\sigma=0.01$ | $\sigma=1.08$ | $\sigma=0.0$ | $\sigma=0.01$ | $\sigma=5.37$ |
| | W64-D2 | 98.7 | 0.94 | 0.96 | 95.3 | 0.47 | 1.19 | 14.7 | 0.47 | 1.44 |
| | | $\sigma=0.0$ | $\sigma=0.19$ | $\sigma=0.83$ | $\sigma=0.0$ | $\sigma=0.01$ | $\sigma=0.86$ | $\sigma=0.0$ | $\sigma=0.01$ | $\sigma=1.00$ |
| | W64-D4 | 98.7 | 0.98 | 1.25 | 98.1 | 0.51 | 1.38 | 24.5 | 0.51 | 3.08 |
| | | $\sigma=0.0$ | $\sigma=0.19$ | $\sigma=1.16$ | $\sigma=0.0$ | $\sigma=0.01$ | $\sigma=1.23$ | $\sigma=0.0$ | $\sigma=0.01$ | $\sigma=4.28$ |
| | W64-D6 | 98.9 | 1.01 | 1.38 | 94.1 | 0.55 | 1.30 | 21.1 | 0.55 | 2.89 |
| | | $\sigma=0.0$ | $\sigma=0.18$ | $\sigma=1.30$ | $\sigma=0.0$ | $\sigma=0.01$ | $\sigma=1.10$ | $\sigma=0.0$ | $\sigma=0.01$ | $\sigma=3.90$ |
| | W128-D2 | 99.9 | 0.93 | 1.40 | 98.3 | 0.48 | 1.43 | 27.3 | 0.48 | 3.91 |
| | | $\sigma=0.0$ | $\sigma=0.19$ | $\sigma=1.32$ | $\sigma=0.0$ | $\sigma=0.01$ | $\sigma=1.25$ | $\sigma=0.0$ | $\sigma=0.01$ | $\sigma=5.96$ |
| | W128-D4 | 98.8 | 0.99 | 1.32 | 94.1 | 0.52 | 1.36 | 21.7 | 0.52 | 2.95 |
| | | $\sigma=0.0$ | $\sigma=0.20$ | $\sigma=1.22$ | $\sigma=0.0$ | $\sigma=0.01$ | $\sigma=1.18$ | $\sigma=0.0$ | $\sigma=0.01$ | $\sigma=3.96$ |
| | W128-D6 | 99.5 | 1.03 | 1.35 | 96.9 | 0.56 | 1.40 | 16.4 | 0.56 | 3.22 |
| | | $\sigma=0.0$ | $\sigma=0.21$ | $\sigma=1.30$ | $\sigma=0.0$ | $\sigma=0.01$ | $\sigma=1.22$ | $\sigma=0.0$ | $\sigma=0.01$ | $\sigma=5.07$ |

Table C.9: Mean and std. dev. for feasibility (%), time (ms), and optimality gap for each ablation: Two Moons.

# D STATEMENT ON USE OF LARGE LANGUAGE MODELS (LLMS)

LLMs did *not* play a significant role in research ideation and/or writing for this paper. Perplexity was used to facilitate literature discovery as part of the research process, alongside other (non-LLM-based) tools. LLMs were also used to aid in sentence-level rewording for a small number of sentences during the preparation of this manuscript.

