# OpenReview forum: "Improving Feasibility via Fast Autoencoder-Based Projections"
_ICLR.cc/2026/Conference — ICLR 2026 Poster_

### Official Review · Reviewer_WGVG · 2025-10-31

**Soundness:** 3
**Presentation:** 2
**Contribution:** 3
**Rating:** 6
**Confidence:** 4

**Summary:**

The paper proposes a FAB projection method, which encodes a prediction and context into a latent space, projects onto a simple convex set 𝑆, and decodes back to the original space. A two-phase training is used. Phase 1 reconstructs the feasible set from feasible samples, while Phase 2 structures the latent via a discriminator with a hinge loss, plus a latent loss and a Jacobian regularizer. FAB is then attached as a plug-and-play mapping 𝜙 around a base network.

**Strengths:**

1. The main contribution is an amortized, non-iterative feasibility mapper that is fast at inference and empirically effective on diverse nonconvex toy sets.
2. Its plug-and-play design is attractive for deployment, since 𝜙 can be mounted onto arbitrary predictors without rewriting training loops.
3. The Safe RL experiment demonstrates the safety and latency advantages.

**Weaknesses:**

1. FAB explicitly does not provide hard feasibility guarantees, making it unsuitable for safety-critical regimes, and no robustness evidence is given for distribution shift or dataset coverage failures that the authors acknowledge as limitations.
2. Comparisons emphasize feasibility, but optimality gaps vary and many classical baselines (e.g., Projected Gradient, FSNet) can reach 100% feasibility as well.
3. The experiments focus on toy nonconvex sets rather than realistic, parameter-dependent instances, so the path to real deployments remains unclear.

**Questions:**

Two biggest concerns with L2O methods remain (1) the expensive training cost, and (2) the lack of feasibility guarantees. Can the authors discuss more about these two aspects?

---

> ### Author Response · Authors · 2025-11-27
> **Author Response to Reviewer WGVG (Part 1/3)**
>
> Thank you to the reviewer for their careful review and thoughtful comments. We appreciate the positive feedback on the latency and empirical safety of the method. We address the identified weaknesses and questions below.
>
> > “FAB explicitly does not provide hard feasibility guarantees, making it unsuitable for safety-critical regimes”
>
> We acknowledge that FAB does not provide formal guarantees typical of exact solvers. Our primary contribution is in providing a data-driven, amortized mechanism for high-speed feasibility enforcement in latency-sensitive, real-time systems. Such a mechanism is helpful in settings where (near-)feasibility is important, but where the benefits of reduced latency outweigh the need for strict constraint satisfaction. This notion of low-latency solutions with prioritized feasibility is helpful across many applications, such as high-frequency robotic control (e.g., quadrotors or locomotion), real-time optimal power flow in energy grids, temperature control of buildings, and high-speed industrial automation. In addition, the method's ability to learn arbitrary sets, as evidenced by the empirical data, makes it useful in situations when no other projection methods are available.
>
> > “No robustness evidence is given for distribution shift or dataset coverage failures that the authors acknowledge as limitations.”
>
> Thank you for raising this important question. While we acknowledge that there is no guarantee that FAB naturally generalizes to unseen constraint layouts, we have now run new experiments to empirically validate this. Specifically, we have conducted two new sets of experiments:  (1) an evaluation on a distinct Safe RL task (SafetyPointPush2-v0), and (2) an ablation study on "incomplete coverage" to simulate domain shift. We have included both of them below and in the updated manuscript.
>
> **1. Generalization to new tasks.**
> To demonstrate that our method handles different constraint dynamics, we extended our evaluation to SafetyGym’s Push2. This task is significantly more challenging than Goal2 as it involves object manipulation (pushing a box), introducing complex contact-based constraints.
>
> | Algorithm | Reward Mean $\uparrow$ | Reward Std $\downarrow$ | Cost Mean $\downarrow$ | Cost Std $\downarrow$ | Time Mean (s) $\downarrow$ |
> | :--- | :---: | :---: | :---: | :---: | :---: |
> | **PPO_FAB** | -0.07 | **0.52** | **7.70** | **37.51** | **3.71** |
> | **PPO** | 0.41 | 3.11 | 59.86 | 120.18 | 3.78 |
> | **PPO_LAG** | **0.60** | 1.58 | 31.34 | 58.17 | 4.09 |
> | **TRPO** | 0.20 | 2.50 | 106.88 | 216.19 | 4.03 |
> | **TRPO_LAG** | -0.77 | 5.51 | 28.22 | 67.21 | 4.05 |
>
> **Table 3:** SafetyGym Push Results
>
> PPO with FAB performed comparably well on this new task, achieving an average cost of 7.70 and a standard deviation of 37.51 across 50 seeds (significantly lower than unconstrained and Lagrangian PPO/TRPO). We do again see that there is a definite “cost to safety” with respect to the reward achieved, as is common in safety-related domains.
>
> It is worth clarifying that, in the context of Safety Gym, “unseen layouts” are handled naturally because FAB operates on the state-action space, which includes ego-centric LiDAR sensors. The autoencoder learns a local safety surrogate (e.g., "if sensor detects obstacle at distance X and velocity is Y, project action to Z"), rather than memorizing a global map. This allows the projection to remain valid across randomized hazard layouts, provided the local sensor-to-constraint dynamics remain consistent.
>
> **2. “Incomplete Coverage” ablations.**
>
> We have completed an ablation study for incomplete coverage of the constraint set in the training data (10%, 25%, 50%, 75% of the constraint set) on the Blob with Bite and Star Shaped constraint families. These results are included in Tables 7 and 8 in Appendix B and below.
>
> In Tables 7 and 8, we provide ablation results for incomplete coverage and decoder capacity. The incomplete coverage ablations test the performance of our method when the training data only consisted of part of the constraint set (10\%, 25\%, 50\%, and 75\%). The decoder capacity ablations test the performance of multiple decoder depth and width configurations. W indicates decoder width, while D indicates decoder depth.
>
> Even with incomplete coverage, the method achieves perfect feasibility across all 18,000 optimization problems we test on these constraint families. For the quadratic and linear problems, the optimality gap is largely unaffected by the training data coverage; for the distance minimization problem, the optimality gap decreases somewhat as coverage increases.

---

> ### Author Response · Authors · 2025-11-27
> **Author Response to Reviewer WGVG (Part 2/3)**
>
> **Table 7:** Mean and std. dev. for feasibility (%), time (ms), and optimality gap for each ablation: Blob with Bite.
>
> | Method | Quad. $\uparrow$ Feas (%) | Quad. $\downarrow$ Time (ms) | Quad. $\downarrow$ Gap | Lin. $\uparrow$ Feas (%) | Lin. $\downarrow$ Time (ms) | Lin. $\downarrow$ Gap | Dist. $\uparrow$ Feas (%) | Dist. $\downarrow$ Time (ms) | Dist. $\downarrow$ Gap |
> | :--- | :---: | :---: | :---: | :---: | :---: | :---: | :---: | :---: | :---: |
> | **Cov-10** | 100.0 | 1.12 | 0.82 | 100.0 | 0.87 | 1.53 | 100.0 | 0.85 | 10.25 |
> | | $\sigma = 0.0$ | $\sigma = 1.10$ | $\sigma = 1.02$ | $\sigma = 0.0$ | $\sigma = 0.04$ | $\sigma = 1.39$ | $\sigma = 0.0$ | $\sigma = 0.04$ | $\sigma = 7.33$ |
> | **Cov-25** | 100.0 | 1.08 | 0.81 | 100.0 | 0.86 | 1.52 | 100.0 | 0.85 | 10.15 |
> | | $\sigma = 0.0$ | $\sigma = 0.09$ | $\sigma = 1.01$ | $\sigma = 0.0$ | $\sigma = 0.01$ | $\sigma = 1.39$ | $\sigma = 0.0$ | $\sigma = 0.03$ | $\sigma = 7.20$ |
> | **Cov-50** | 100.0 | 1.08 | 0.80 | 100.0 | 0.86 | 1.47 | 100.0 | 0.85 | 9.15 |
> | | $\sigma = 0.0$ | $\sigma = 0.09$ | $\sigma = 0.98$ | $\sigma = 0.0$ | $\sigma = 0.01$ | $\sigma = 1.30$ | $\sigma = 0.0$ | $\sigma = 0.03$ | $\sigma = 6.97$ |
> | **Cov-75** | 100.0 | 1.08 | 0.81 | 100.0 | 0.86 | 1.52 | 100.0 | 0.85 | 9.46 |
> | | $\sigma = 0.0$ | $\sigma = 0.09$ | $\sigma = 1.01$ | $\sigma = 0.0$ | $\sigma = 0.03$ | $\sigma = 1.38$ | $\sigma = 0.0$ | $\sigma = 0.03$ | $\sigma = 6.86$ |
>
>
> **Table 8:** Mean and std. dev. for feasibility (%), time (ms), and optimality gap for each ablation: Star Shaped.
>
> | Method | Quad. $\uparrow$ Feas (%) | Quad. $\downarrow$ Time (ms) | Quad. $\downarrow$ Gap | Lin. $\uparrow$ Feas (%) | Lin. $\downarrow$ Time (ms) | Lin. $\downarrow$ Gap | Dist. $\uparrow$ Feas (%) | Dist. $\downarrow$ Time (ms) | Dist. $\downarrow$ Gap |
> | :--- | :---: | :---: | :---: | :---: | :---: | :---: | :---: | :---: | :---: |
> | **Cov-10** | 100.0 | 1.06 | 1.30 | 100.0 | 0.87 | 1.43 | 100.0 | 0.88 | 9.57 |
> | | $\sigma = 0.0$ | $\sigma = 0.11$ | $\sigma = 1.25$ | $\sigma = 0.0$ | $\sigma = 0.02$ | $\sigma = 1.40$ | $\sigma = 0.0$ | $\sigma = 0.03$ | $\sigma = 7.80$ |
> | **Cov-25** | 100.0 | 1.04 | 1.21 | 100.0 | 0.86 | 1.00 | 100.0 | 0.87 | 5.31 |
> | | $\sigma = 0.0$ | $\sigma = 0.09$ | $\sigma = 1.36$ | $\sigma = 0.0$ | $\sigma = 0.01$ | $\sigma = 0.99$ | $\sigma = 0.0$ | $\sigma = 0.02$ | $\sigma = 4.25$ |
> | **Cov-50** | 100.0 | 1.05 | 0.84 | 100.0 | 0.86 | 0.92 | 100.0 | 0.87 | 5.93 |
> | | $\sigma = 0.0$ | $\sigma = 0.09$ | $\sigma = 0.93$ | $\sigma = 0.0$ | $\sigma = 0.01$ | $\sigma = 0.91$ | $\sigma = 0.0$ | $\sigma = 0.03$ | $\sigma = 5.43$ |
> | **Cov-75** | 100.0 | 1.05 | 0.75 | 100.0 | 0.86 | 0.82 | 100.0 | 0.87 | 4.72 |
> | | $\sigma = 0.0$ | $\sigma = 0.09$ | $\sigma = 0.78$ | $\sigma = 0.0$ | $\sigma = 0.01$ | $\sigma = 0.75$ | $\sigma = 0.0$ | $\sigma = 0.03$ | $\sigma = 4.39$ |
>
>
> > “Comparisons emphasize feasibility, but optimality gaps vary and many classical baselines (e.g., Projected Gradient, FSNet) can reach 100% feasibility as well.”
>
> We agree that classical baselines can achieve high feasibility in certain regimes. However, we believe our method offers a distinct and critical advantage in terms of latency/speed and empirical robustness to complex, non-convex constraints, which are the primary motivations for this work.
>
> The primary contribution of FAB is providing a high-fidelity projection at a fraction of the computational cost of iterative methods. As shown in Tables 1, 4, 5, and 6, existing methods typically require 20-200ms per instance due to their iterative nature. In contrast, FAB operates as a direct feedforward pass, consistently achieving sub-millisecond (<1ms) inference times. In addition, FAB (specifically its multi-decoder implementations) achieve 100% feasibility on disjoint, non-convex sets, such as Two Moons (Table 4), while maintaining <1ms inference times.
>
> Regarding optimality, the optimality gap often depends on the difficulty of the optimization problem. Empirically, FAB still achieves small optimality gaps compared to other baselines (Tables 1, 4, 5, 6).

---

> ### Author Response · Authors · 2025-11-27
> **Author Response to Reviewer WGVG (Part 3/3)**
>
> > “The experiments focus on toy nonconvex sets rather than realistic, parameter-dependent instances, so the path to real deployments remains unclear.”
>
> Thank you for the comment.
>
> First, we have added a clarification in Section 5 that the Safe RL tasks we investigate are parameter-dependent. As defined in Section 5, the constraint set $\mathcal{C}(s_k)$ is parameter-dependent, as the feasible set of actions changes dynamically at every timestep $k$ based on the agent’s high-dimensional state vector $s_k$ (the parameter). The agent then tries to satisfy these changing constraints (e.g. dynamic distance to hazards) in real time.
>
> To further demonstrate that FAB handles complex, realistic, parameter-dependent dynamics, we extended our evaluation to SafetyGym’s Push2 task, which is more challenging than the Goal2 task. (Results shown above.)
>
> Finally, regarding the path to real deployments, we argue that FAB becomes increasingly beneficial as scale and constraint complexity increase. The toy nonconvex sets in Section 4 were included specifically to stress-test the method against topological challenges (e.g., disjoint sets) that traditional methods do not handle well, serving as a necessary precursor to these dynamic control tasks. In realistic deployments (e.g., high-frequency robotics), exact constraint enforcement via iterative solvers can create a computational bottleneck that scales poorly with the complexity of the constraints. Therefore, the "path to deployment" for FAB is specifically advantageous in regimes where constraints are too complex for real-time solvers to handle within the control frequency.

---

> > ### Author Response · Authors · 2025-12-03
> > **“Incomplete Coverage” ablations (Continued)**
> >
> > We are following up to share additional results as part of the “Incomplete Coverage” ablations, for the Two Moons constraint family.
> >
> > | Method | Feasibility rate (±std) | Time (ms) (±std) | Opt. Gap (±std) |
> > | :--- | :---: | :---: | :---: |
> > | Cov-10 | 100.0% ± 0.0% | 1.48 ± 1.05 | 1.6433 ± 1.5002 |
> > | Cov-25 | 76.5% ± 24.7% | 1.46 ± 0.15 | 2.1547 ± 1.9190 |
> > | Cov-50 | 92.7% ± 26.0% | 1.45 ± 0.11 | 0.8246 ± 0.8971 |
> > | Cov-75 | 91.2% ± 28.3% | 1.46 ± 0.13 | 0.9018 ± 1.0153 |
> > **Two Moons: Quadratic objective with incomplete coverage (10%, 25%, 50%, 75%)**
> >
> > We are currently finishing the remaining ablations and plan to include them in the final paper as soon as possible. Thank you for your understanding!

---

### Official Review · Reviewer_nF3p · 2025-11-02

**Soundness:** 3
**Presentation:** 3
**Contribution:** 2
**Rating:** 6
**Confidence:** 2

**Summary:**

The paper proposes FAB, a fast “approximate projector” built from a conditional autoencoder. Training has two phases: (1) reconstruct feasible points; (2) use a discriminator + simple penalties to make latent samples decode to feasible outputs. Experiments on several synthetic constraint families and one safe-RL task look promising.

**Strengths:**

1. Easy to attach after an existing predictor to improve feasibility with negligible latency.

2. The two-phase training architecture is clear and easy to follow

3. Tables report near-perfect feasibility and big time gains vs. homeomorphic projection and other baselines

**Weaknesses:**

It would further strengthen the paper if some form of theoretical guarantee—even a weak or probabilistic feasibility bound—could be established.

**Questions:**

1. Could you report throughput vs. dimension and memory for different decoder widths and number of decoders, and a cost breakdown?

2. Does a FAB trained on one hazard layout transfer to unseen layouts? Any results under domain shift?

---

> ### Author Response · Authors · 2025-11-27
> **Author Response to Reviewer nF3p (Part 1/3)**
>
> Thank you to the reviewer for their careful review and thoughtful comments. We appreciate the positive feedback on the clarity of presentation and the acknowledgement of the method’s strengths. We address the identified weaknesses and questions below.
>
> > “Could you report throughput vs. dimension and memory for different decoder widths and number of decoders, and a cost breakdown?”
>
> We are currently running the full suite of benchmarks regarding throughput vs. dimension, memory profiling, and the specific cost breakdown. We will update the thread with those results as soon as the experiments conclude.
>
> At the moment, we have completed an ablation study for decoder capacity (varying width W and depth D) on the Blob with Bite and Star Shaped constraint families. We have included these results in Tables 7 and 8 in Appendix B, as well as below.
>
> In Tables 7 and 8, we provide ablation results for incomplete coverage and decoder capacity. The incomplete coverage ablations test the performance of our method when the training data consisted only of part of the constraint set (10\%, 25\%, 50\%, and 75\%). The decoder capacity ablations test the performance of multiple decoder depth and width configurations. W indicates decoder width, while D indicates decoder depth.
>
> | Method | Quad. $\uparrow$ Feas (%) | Quad. $\downarrow$ Time (ms) | Quad. $\downarrow$ Gap | Lin. $\uparrow$ Feas (%) | Lin. $\downarrow$ Time (ms) | Lin. $\downarrow$ Gap | Dist. $\uparrow$ Feas (%) | Dist. $\downarrow$ Time (ms) | Dist. $\downarrow$ Gap |
> | :--- | :---: | :---: | :---: | :---: | :---: | :---: | :---: | :---: | :---: |
> | **Cov-10** | 100.0 | 1.12 | 0.82 | 100.0 | 0.87 | 1.53 | 100.0 | 0.85 | 10.25 |
> | | $\sigma = 0.0$ | $\sigma = 1.10$ | $\sigma = 1.02$ | $\sigma = 0.0$ | $\sigma = 0.04$ | $\sigma = 1.39$ | $\sigma = 0.0$ | $\sigma = 0.04$ | $\sigma = 7.33$ |
> | **Cov-25** | 100.0 | 1.08 | 0.81 | 100.0 | 0.86 | 1.52 | 100.0 | 0.85 | 10.15 |
> | | $\sigma = 0.0$ | $\sigma = 0.09$ | $\sigma = 1.01$ | $\sigma = 0.0$ | $\sigma = 0.01$ | $\sigma = 1.39$ | $\sigma = 0.0$ | $\sigma = 0.03$ | $\sigma = 7.20$ |
> | **Cov-50** | 100.0 | 1.08 | 0.80 | 100.0 | 0.86 | 1.47 | 100.0 | 0.85 | 9.15 |
> | | $\sigma = 0.0$ | $\sigma = 0.09$ | $\sigma = 0.98$ | $\sigma = 0.0$ | $\sigma = 0.01$ | $\sigma = 1.30$ | $\sigma = 0.0$ | $\sigma = 0.03$ | $\sigma = 6.97$ |
> | **Cov-75** | 100.0 | 1.08 | 0.81 | 100.0 | 0.86 | 1.52 | 100.0 | 0.85 | 9.46 |
> | | $\sigma = 0.0$ | $\sigma = 0.09$ | $\sigma = 1.01$ | $\sigma = 0.0$ | $\sigma = 0.03$ | $\sigma = 1.38$ | $\sigma = 0.0$ | $\sigma = 0.03$ | $\sigma = 6.86$ |
> | **W32-D2** | 99.5 | 1.09 | 0.62 | 98.5 | 0.79 | 1.21 | 72.5 | 0.80 | 6.36 |
> | | $\sigma = 6.8$ | $\sigma = 1.48$ | $\sigma = 0.80$ | $\sigma = 12.0$ | $\sigma = 0.03$ | $\sigma = 1.03$ | $\sigma = 44.6$ | $\sigma = 0.03$ | $\sigma = 5.14$ |
> | **W32-D4** | 98.6 | 1.11 | 0.67 | 98.5 | 0.86 | 1.35 | 77.3 | 0.88 | 9.52 |
> | | $\sigma = 11.8$ | $\sigma = 0.08$ | $\sigma = 0.88$ | $\sigma = 12.3$ | $\sigma = 0.03$ | $\sigma = 1.23$ | $\sigma = 41.9$ | $\sigma = 0.03$ | $\sigma = 7.37$ |
> | **W32-D6** | 99.4 | 1.17 | 0.72 | 98.7 | 0.93 | 1.20 | 96.8 | 0.95 | 6.54 |
> | | $\sigma = 7.7$ | $\sigma = 0.08$ | $\sigma = 0.81$ | $\sigma = 11.5$ | $\sigma = 0.03$ | $\sigma = 1.06$ | $\sigma = 17.6$ | $\sigma = 0.03$ | $\sigma = 5.17$ |
> | **W64-D2** | 100.0 | 1.03 | 0.58 | 100.0 | 0.79 | 1.17 | 100.0 | 0.80 | 6.19 |
> | | $\sigma = 0.0$ | $\sigma = 0.07$ | $\sigma = 0.77$ | $\sigma = 0.0$ | $\sigma = 0.03$ | $\sigma = 0.99$ | $\sigma = 0.0$ | $\sigma = 0.02$ | $\sigma = 5.09$ |
> | **W64-D6** | 67.3 | 1.16 | 0.76 | 68.6 | 0.93 | 1.42 | 52.2 | 0.94 | 7.10 |
> | | $\sigma = 46.9$ | $\sigma = 0.08$ | $\sigma = 0.93$ | $\sigma = 46.4$ | $\sigma = 0.03$ | $\sigma = 1.28$ | $\sigma = 50.0$ | $\sigma = 0.02$ | $\sigma = 5.50$ |
> | **W128-D2** | 99.9 | 1.04 | 0.57 | 99.7 | 0.79 | 1.22 | 95.3 | 0.80 | 6.13 |
> | | $\sigma = 3.7$ | $\sigma = 0.07$ | $\sigma = 0.76$ | $\sigma = 5.2$ | $\sigma = 0.03$ | $\sigma = 1.03$ | $\sigma = 21.2$ | $\sigma = 0.02$ | $\sigma = 5.08$ |
> | **W128-D4** | 100.0 | 1.10 | 0.82 | 100.0 | 0.86 | 1.51 | 100.0 | 0.87 | 9.70 |
> | | $\sigma = 0.0$ | $\sigma = 0.07$ | $\sigma = 0.98$ | $\sigma = 0.0$ | $\sigma = 0.03$ | $\sigma = 1.36$ | $\sigma = 0.0$ | $\sigma = 0.02$ | $\sigma = 7.01$ |
> | **W128-D6** | 100.0 | 1.16 | 0.82 | 100.0 | 0.93 | 1.55 | 100.0 | 0.94 | 10.26 |
> | | $\sigma = 0.0$ | $\sigma = 0.08$ | $\sigma = 1.02$ | $\sigma = 0.0$ | $\sigma = 0.03$ | $\sigma = 1.41$ | $\sigma = 0.0$ | $\sigma = 0.06$ | $\sigma = 7.38$ |
>
> **Table 7:** Mean and std. dev. for feasibility (%), time (ms), and optimality gap for each ablation: Blob with Bite.

---

> ### Author Response · Authors · 2025-11-27
> **Author Response to Reviewer nF3p (Part 2/3)**
>
> | Method | Quad. $\uparrow$ Feas (%) | Quad. $\downarrow$ Time (ms) | Quad. $\downarrow$ Gap | Lin. $\uparrow$ Feas (%) | Lin. $\downarrow$ Time (ms) | Lin. $\downarrow$ Gap | Dist. $\uparrow$ Feas (%) | Dist. $\downarrow$ Time (ms) | Dist. $\downarrow$ Gap |
> | :--- | :---: | :---: | :---: | :---: | :---: | :---: | :---: | :---: | :---: |
> | **Cov-10** | 100.0 | 1.06 | 1.30 | 100.0 | 0.87 | 1.43 | 100.0 | 0.88 | 9.57 |
> | | $\sigma = 0.0$ | $\sigma = 0.11$ | $\sigma = 1.25$ | $\sigma = 0.0$ | $\sigma = 0.02$ | $\sigma = 1.40$ | $\sigma = 0.0$ | $\sigma = 0.03$ | $\sigma = 7.80$ |
> | **Cov-25** | 100.0 | 1.04 | 1.21 | 100.0 | 0.86 | 1.00 | 100.0 | 0.87 | 5.31 |
> | | $\sigma = 0.0$ | $\sigma = 0.09$ | $\sigma = 1.36$ | $\sigma = 0.0$ | $\sigma = 0.01$ | $\sigma = 0.99$ | $\sigma = 0.0$ | $\sigma = 0.02$ | $\sigma = 4.25$ |
> | **Cov-50** | 100.0 | 1.05 | 0.84 | 100.0 | 0.86 | 0.92 | 100.0 | 0.87 | 5.93 |
> | | $\sigma = 0.0$ | $\sigma = 0.09$ | $\sigma = 0.93$ | $\sigma = 0.0$ | $\sigma = 0.01$ | $\sigma = 0.91$ | $\sigma = 0.0$ | $\sigma = 0.03$ | $\sigma = 5.43$ |
> | **Cov-75** | 100.0 | 1.05 | 0.75 | 100.0 | 0.86 | 0.82 | 100.0 | 0.87 | 4.72 |
> | | $\sigma = 0.0$ | $\sigma = 0.09$ | $\sigma = 0.78$ | $\sigma = 0.0$ | $\sigma = 0.01$ | $\sigma = 0.75$ | $\sigma = 0.0$ | $\sigma = 0.03$ | $\sigma = 4.39$ |
> | **W32-D2** | 100.0 | 1.01 | 0.41 | 100.0 | 0.81 | 0.76 | 100.0 | 0.81 | 3.66 |
> | | $\sigma = 0.0$ | $\sigma = 0.11$ | $\sigma = 0.48$ | $\sigma = 0.0$ | $\sigma = 0.03$ | $\sigma = 0.66$ | $\sigma = 0.0$ | $\sigma = 0.03$ | $\sigma = 3.48$ |
> | **W32-D4** | 100.0 | 1.07 | 0.37 | 100.0 | 0.88 | 0.79 | 100.0 | 0.88 | 3.55 |
> | | $\sigma = 0.0$ | $\sigma = 0.09$ | $\sigma = 0.44$ | $\sigma = 0.0$ | $\sigma = 0.05$ | $\sigma = 0.68$ | $\sigma = 0.0$ | $\sigma = 0.04$ | $\sigma = 3.46$ |
> | **W32-D6** | 100.0 | 1.13 | 0.39 | 100.0 | 0.95 | 0.77 | 100.0 | 0.95 | 3.77 |
> | | $\sigma = 0.0$ | $\sigma = 0.09$ | $\sigma = 0.46$ | $\sigma = 0.0$ | $\sigma = 0.03$ | $\sigma = 0.66$ | $\sigma = 0.0$ | $\sigma = 0.02$ | $\sigma = 3.58$ |
> | **W64-D2** | 100.0 | 1.00 | 0.39 | 100.0 | 0.80 | 0.76 | 100.0 | 0.81 | 3.74 |
> | | $\sigma = 0.0$ | $\sigma = 0.09$ | $\sigma = 0.46$ | $\sigma = 0.0$ | $\sigma = 0.02$ | $\sigma = 0.65$ | $\sigma = 0.0$ | $\sigma = 0.02$ | $\sigma = 3.54$ |
> | **W64-D6** | 100.0 | 1.14 | 0.87 | 100.0 | 0.95 | 1.06 | 100.0 | 0.95 | 7.90 |
> | | $\sigma = 0.0$ | $\sigma = 0.09$ | $\sigma = 1.10$ | $\sigma = 0.0$ | $\sigma = 0.03$ | $\sigma = 1.11$ | $\sigma = 0.0$ | $\sigma = 0.02$ | $\sigma = 7.51$ |
> | **W128-D2** | 100.0 | 0.99 | 0.38 | 100.0 | 0.81 | 0.77 | 100.0 | 0.81 | 3.62 |
> | | $\sigma = 0.0$ | $\sigma = 0.09$ | $\sigma = 0.44$ | $\sigma = 0.0$ | $\sigma = 0.02$ | $\sigma = 0.67$ | $\sigma = 0.0$ | $\sigma = 0.02$ | $\sigma = 3.48$ |
> | **W128-D4** | 100.0 | 1.05 | 0.38 | 100.0 | 0.88 | 0.83 | 100.0 | 0.88 | 3.51 |
> | | $\sigma = 0.0$ | $\sigma = 0.09$ | $\sigma = 0.44$ | $\sigma = 0.0$ | $\sigma = 0.01$ | $\sigma = 0.73$ | $\sigma = 0.0$ | $\sigma = 0.02$ | $\sigma = 3.34$ |
> | **W128-D6** | 100.0 | 1.14 | 0.37 | 100.0 | 0.96 | 0.74 | 100.0 | 0.95 | 3.40 |
> | | $\sigma = 0.0$ | $\sigma = 0.09$ | $\sigma = 0.44$ | $\sigma = 0.0$ | $\sigma = 0.01$ | $\sigma = 0.63$ | $\sigma = 0.0$ | $\sigma = 0.02$ | $\sigma = 3.39$ |
>
> **Table 8:** Mean and std. dev. for feasibility (%), time (ms), and optimality gap for each ablation: Star Shaped.
>
> The main takeaway is that increasing the decoder capacity from Width-32/Depth-2 up to Width-128/Depth-6 results in only a marginal increase in inference time (from ~1.01ms to ~1.14ms). This suggests that the network size may not significantly impact inference time at this scale.

---

> ### Author Response · Authors · 2025-11-27
> **Author Response to Reviewer nF3p (Part 3/3)**
>
> > “Does a FAB trained on one hazard layout transfer to unseen layouts? Any results under domain shift?”
>
> Thank you for raising this important question. While we acknowledge that there is no guarantee that FAB naturally generalizes to unseen constraint layouts, we have now run new experiments to empirically validate this. Specifically, we have conducted two new sets of experiments:  (1) an evaluation on a distinct Safe RL task (SafetyPointPush2-v0), and (2) an ablation study on "incomplete coverage" to simulate domain shift. We have included both of them below and in the updated manuscript.
>
> **1. Generalization to new tasks.**
> To demonstrate that our method handles different constraint dynamics, we extended our evaluation to SafetyGym’s Push2. This task is significantly more challenging than Goal2 as it involves object manipulation (pushing a box), introducing complex contact-based constraints.
>
> | Algorithm | Reward Mean $\uparrow$ | Reward Std $\downarrow$ | Cost Mean $\downarrow$ | Cost Std $\downarrow$ | Time Mean (s) $\downarrow$ |
> | :--- | :---: | :---: | :---: | :---: | :---: |
> | **PPO_FAB** | -0.07 | **0.52** | **7.70** | **37.51** | **3.71** |
> | **PPO** | 0.41 | 3.11 | 59.86 | 120.18 | 3.78 |
> | **PPO_LAG** | **0.60** | 1.58 | 31.34 | 58.17 | 4.09 |
> | **TRPO** | 0.20 | 2.50 | 106.88 | 216.19 | 4.03 |
> | **TRPO_LAG** | -0.77 | 5.51 | 28.22 | 67.21 | 4.05 |
>
> **Table 3:** SafetyGym Push Results
>
> PPO with FAB performed comparably well on this new task, achieving an average cost of 7.70 and a standard deviation of 37.51 across 50 seeds (significantly lower than unconstrained and Lagrangian PPO/TRPO). We again see a definite “cost to safety” relative to the reward achieved, as is common in safety-related domains.
>
> It is worth clarifying that, in the context of Safety Gym, “unseen layouts” are handled naturally because FAB operates on the state-action space, which includes ego-centric LiDAR sensors. The autoencoder learns a local safety surrogate (e.g., "if sensor detects obstacle at distance X and velocity is Y, project action to Z"), rather than memorizing a global map. This allows the projection to remain valid across randomized hazard layouts, provided the local sensor-to-constraint dynamics remain consistent.
>
> **2. "Incomplete Coverage" ablations.**
>
> We have completed an ablation study for incomplete coverage of the constraint set in the training data (10%, 25%, 50%, 75% of the constraint set) on the Blob with Bite and Star Shaped constraint families. These results are included in Tables 7 and 8 in Appendix B and above.
>
> Even with incomplete coverage, the method achieves perfect feasibility across all 18,000 optimization problems we test on these constraint families. For the quadratic and linear problems, the optimality gap is largely unaffected by the training data coverage; for the distance minimization problem, the optimality gap decreases somewhat as coverage increases.

---

> > ### Author Response · Authors · 2025-12-03
> > **“Incomplete Coverage” ablations (continued)**
> >
> > > “Does a FAB trained on one hazard layout transfer to unseen layouts? Any results under domain shift?”
> >
> > We are following up to share additional results as part of the “Incomplete Coverage” ablations, for the Two Moons constraint family.
> >
> > | Method | Feasibility rate (±std) | Time (ms) (±std) | Opt. Gap (±std) |
> > | :--- | :---: | :---: | :---: |
> > | Cov-10 | 100.0% ± 0.0% | 1.48 ± 1.05 | 1.6433 ± 1.5002 |
> > | Cov-25 | 76.5% ± 24.7% | 1.46 ± 0.15 | 2.1547 ± 1.9190 |
> > | Cov-50 | 92.7% ± 26.0% | 1.45 ± 0.11 | 0.8246 ± 0.8971 |
> > | Cov-75 | 91.2% ± 28.3% | 1.46 ± 0.13 | 0.9018 ± 1.0153 |
> > **Two Moons: Quadratic objective with incomplete coverage (10%, 25%, 50%, 75%)**
> >
> > We are currently finishing the remaining ablations and plan to include them in the final paper as soon as possible. Thank you for your understanding!

---

### Official Review · Reviewer_3AaQ · 2025-11-07

**Soundness:** 3
**Presentation:** 3
**Contribution:** 2
**Rating:** 4
**Confidence:** 5

**Summary:**

This paper introduces the Feasibility Autoencoder-Based (FAB) method, which uses an offline-trained autoencoder to project infeasible DNN outputs into a structured latent space for fast feasibility enforcement in constrained optimization. Key contributions include a two-phase training algorithm for the autoencoder, empirical speedups, and demonstrations of high feasibility with near-optimal performance on benchmarks like portfolio optimization.

**Strengths:**

1. The method's originality lies in its pretrained encoder-decoder as a reusable feasibility projector, combining autoencoders with adversarial structuring for non-convex constraints and showing potential for generalization across tasks.

2. Quality is good, with well-designed algorithms, and good presentation.

3. The studied problem is important, as FAB enables real-time constraint handling in practical ML applications, potentially impacting fields like operations research and safe AI.

**Weaknesses:**

1. The lack of theoretical guarantees, especially on optimality preservation post-projection, weakens reliability, as the method relies on empirical data without bounds on gaps or failure modes. The feasibility and optimality guarantee is not presented and the method itself seems only rely on the performance of the decoder

2. Contributions are vague compared to baselines like homeomorphic projections, which offer low-complexity feasibility guarantees; clearer differentiation (e.g., quantitative edges in non-convex settings) is needed.

3. The rationale of optimality consideration is questionable, please see below.

**Questions:**

1. How does the autoencoder ensure optimality conservation during projection, and can you provide bounds or discuss failure modes?

2. How does training handle multi-lable T_feas (could have multiple y per x)? Will different y harms the model performance as it could be confused on the targeted y.

3. What are FAB's specific advantages over homeomorphic projections (e.g., in non-convexity or generalization), and how could the pretrained encoder-decoder be extended to new domains?

4. The rationale of optimality consideration is questionable, like the decoder only see feaible/infeasible points, given an infeasible but close to optimal point, how could the model reconstruct a good solution with good optimality performance?

---

> ### Author Response · Authors · 2025-11-27
> **Author Response to Reviewer 3AaQ (Part 1/2)**
>
> Thank you to the reviewer for their thoughtful comments. We appreciate the positive feedback on the method's originality and the clarity of presentation. We address the identified weaknesses and questions below.
>
> # 1. Optimality guarantees and reliability.
>
> In deep learning-based constrained optimization, the general “template” of the methods is that they involve a neural network ($N_{\theta}$) followed by a projection/correction step ($\phi_x$). The neural network plus projection/correction are then trained end-to-end in order to improve optimality. That is, the feasibility correction alone is not aiming to achieve optimality – that is the role of the end-to-end training pipeline, which trains on an optimality-based loss. To the best of our knowledge, no methods in this class of methods provide broad, problem-independent optimality guarantees; existing guarantees are limited to $\varepsilon$-optimality under restrictive assumptions on the problem class or architecture [1]. However, via end-to-end neural network training, they aim to achieve good (empirical) optimality performance while maintaining (empirical) feasibility.
>
> Our work provides a data-driven alternative to the feasibility correction ($\phi_x$) mechanism in the above methods, which is fast to run and (like the other feasibility correction procedures) can be used as part of an end-to-end training pipeline. It is meant to fill a gap in situations where exact feasibility is not needed or too expensive, and it is not intended to replace methods with provable feasibility guarantees in situations where that is needed. This notion of low-latency solutions with prioritized feasibility is helpful across many applications, such as high-frequency robotic control (e.g., quadrotors or locomotion), real-time optimal power flow in energy grids, temperature control of buildings, and high-speed industrial automation.
>
> [1] E. Liang, M. Chen, and S. H. Low, "Homeomorphic projection to ensure neural-network solution feasibility for constrained optimization," Journal of Machine Learning Research, vol. 25, no. 329, pp. 1–55, 2024.
>
> > “The lack of theoretical guarantees, especially on optimality preservation post-projection, weakens reliability, as the method relies on empirical data without bounds on gaps or failure modes”
>
> We acknowledge that FAB does not provide formal guarantees typical of exact solvers. Our primary contribution is in providing a data-driven, amortized mechanism for high-speed feasibility enforcement in latency-sensitive, real-time systems. Such a mechanism is helpful in settings where (near-)feasibility is important, but where the benefits of reduced latency outweigh the need for strict constraint satisfaction. In addition, the method's ability to learn arbitrary sets, as evidenced by the empirical data, makes it useful in situations when no other projection methods are available.
>
> > “The feasibility and optimality guarantee is not presented and the method itself seems only rely on the performance of the decoder”
> > “How does the autoencoder ensure optimality conservation during projection, and can you provide bounds or discuss failure modes?”
>
> As mentioned above, the autoencoder projection itself is not responsible for optimality, but only for improving feasibility; instead, the solution network $N_{\theta}$ is trained end-to-end with the trained autoencoder $\phi_x$ to aim to output empirically feasible, optimal solutions to the optimization problem. However, it is a valid concern that the autoencoder could provide a distorted feasibility correction in some cases (e.g., in out-of-distribution cases) that could impede overall optimality at inference. To address this concern, low-distortion incentives are built into the training process. One of the objective terms in Phase 2 is the Jacobian regularization term (Eq. 9), which encourages uniform coverage of the feasible set by the decoder and low distortion. This means that if $N_{\theta}$ outputs a particular point, the projection is designed to move the point minimally while enforcing feasibility. The empirical data across the constrained optimization problems provide supporting evidence that the optimality gaps for FAB and its ablations are comparable to those of baseline methods.

---

> ### Author Response · Authors · 2025-11-27
> **Author Response to Reviewer 3AaQ (Part 2/2)**
>
> > “The rationale of optimality consideration is questionable, like the decoder only see feasible/infeasible points, given an infeasible but close to optimal point, how could the model reconstruct a good solution with good optimality performance?”
>
> We have refined the description in Section 3.2 to clarify the role of each component in the pipeline:
> - The purpose of the autoencoder $\phi_x$ is purely projection, not optimization. It is trained offline using labelled feasibility data in order to learn the geometry of the feasible set (the feasibility mapping).
> - The solution network $N_{\theta}$ is used for the optimization. It is trained on problem parameters and solutions to output optimal solutions.
>
> In addition, one of the objective terms in Phase 2 is the Jacobian regularization term (Eq. 9), which encourages uniform coverage of the feasible set by the decoder and low distortion. Therefore, as mentioned above, the model is incentivized to reconstruct feasible points as accurately as possible.
>
> # 2. Handling multi-label T_feas.
> > “How does training handle multi-label T_feas (could have multiple y per x)? Will different y harms the model performance as it could be confused on the targeted y.”
>
> The autoencoder learns a deterministic mapping from the input to the output. It is trained to reconstruct any feasible point $y$ provided in the input, learning the boundaries and structure of the feasible set. To account for optimality, $N_{\theta}$ learns to adapt to the autoencoder’s outputs as part of the end-to-end training. Then, during inference, the upstream network $N_{\theta}$ is responsible for selecting which specific optimal point to target. Therefore, the existence of multiple outputs $y$ per $x$ does not confuse the system, as $N_{\theta}$ adapts to select the most optimal solution or a single optimal point. We have now clarified this point further in the paper (Section 3.2).
>
> # 3. Advantages over existing methods.
> > “What are FAB's specific advantages over homeomorphic projections (e.g., in non-convexity or generalization)?”
>
> We have sharpened the distinction between the methods in Section 2 (Related Work) and added more results related to the Homeomorphic Projection (Tables 4 and 6). Our method offers several noticeable advantages, primarily:
> 1. **FAB projections are fast and one-shot**, achieving sub-1ms inference. For instance, on the Star-Shaped family (Table 4), FAB achieves a mean time of ~0.6 ms, compared to ~127 ms for the Homeomorphic Projection baseline, which relies on the iterative bisection projection operation. This makes FAB projections suitable for latency-sensitive applications.
> 2. **Our work presents a method that can learn a mapping between a latent hypersphere and any continuous constraint set**, and is compatible with end-to-end training and inference. The framework is very general. In contrast, Homeomorphic Projections (HP) are mostly only suitable for ball-homeomorphic sets, limiting their applicability across domains. We’ve added more HP results to Tables 4 and 6 (the Two Moons and Concentric Circles constraint families). The feasibility rates decrease noticeably as these constraint sets are non-ball-homeomorphic. Meanwhile, FAB successfully handles these cases, as well as the two parameter-dependent safe RL tasks (as demonstrated in Tables 2, 3, and 5).
> 3. **The pretrained autoencoder is inherently plug-and-play and compatible with end-to-end training**, while HP offers post-hoc feasibility correction.
>
> > “How could the pretrained encoder-decoder be extended to new domains?”
>
> To extend the pretrained encoder-decoder to new domains, we would collect a labeled set of feasible and infeasible points and execute the procedure outlined in the paper. If the new constraints share structural similarities with previously learned constraints, the autoencoder weights can be fine-tuned rather than trained from scratch, potentially improving sample efficiency. While the requirement to retrain is a trade-off compared to generic solvers, it is precisely this offline amortized learning that enables massive gains in online inference speed.

---

### Author Response · Authors · 2025-12-03
**Summary of the Rebuttal Process**

Many thanks to the previous AC, the newly-assigned AC, and the reviewers for their time and thoughtful engagement with our work. Following the conference’s recent guidance, we provide here a concise overview of the revisions made in direct response to reviewers' feedback.
During the rebuttal period, we revised the manuscript with new experiments and ablations, and made various clarity improvements in response to the reviewer feedback. Specifically, we:

- *Added two new experiments to further assess the impact of distribution shift*: (1) evaluation on a distinct Safe RL task (SafetyPointPush2-v0) in addition to SafetyPointGoal2-v0 in the original submission, and (2) an ablation study on "incomplete coverage" in the autoencoder training set to simulate domain shift. In SafetyPointPush2-v0, we see that our method successfully reduces constraint violations. Likewise, across various “incomplete coverage” settings, our method perfectly enforces feasibility in our amortized optimization experimental settings. Together, these experiments indicate that our method is, indeed, able to handle distribution shifts.
- *Added tests of the effect of different decoder depth and width configurations in the constrained optimization settings*. The main takeaway is that increasing the decoder capacity from Width-32/Depth-2 up to Width-128/Depth-6 results in only a marginal increase in inference time (from ~1.01ms to ~1.14ms), suggesting that network size may not significantly impact inference time at this scale.
- *Clarified the role of the autoencoder vs. the end-to-end training pipeline*, where the autoencoder $\phi_x$ is responsible for (approximate) feasibility, whereas the overall neural network plus approximate projection $N_{\theta} \circ \phi_x$ is trained end-to-end to improve optimality.
- *Further clarified the advantages of our method over homeomorphic projection*, notably in computational speed and the ability to handle arbitrary continuous constraint sets.

While reviewers were not able to engage directly with our responses due to the changed review process, we believe we have successfully addressed all their concerns. Thank you again to the ACs and reviewers for their effort and engagement!

---

### Meta-Review · Area_Chair_sqU1 · 2026-01-08

**Summary:**

The reviewers acknowledged the paper’s strengths: a novel feasibility autoencoder (FAB) for fast, amortized projection onto non‑convex sets, clear two‑phase training. However, significant concerns were raised that informed the initial scores (one borderline‑negative 4, two borderline‑positive 6s). These concerns centered on:

No formal bounds on optimality preservation or feasibility, especially under distribution shift.

Limited evidence of generalization to unseen constraint layouts or parameter‑dependent settings.

Insufficient differentiation from baselines like homeomorphic projections, and questions about the optimality rationale.

**Reviewer Concerns:**

Addressed Concerns:

New experiments on a distinct Safe‑RL task (SafetyPointPush2‑v0) and “incomplete‑coverage” ablations (10–75% of the constraint set) show that FAB maintains high feasibility even with limited training data, demonstrating robustness to distribution shift.

The rebuttal sharpens the comparison with homeomorphic projections, highlighting FAB’s speed (sub‑1ms vs. iterative baselines) and ability to handle non‑ball‑homeomorphic sets (e.g., Two Moons).

Outstanding Concerns:

The authors acknowledge FAB does not provide hard feasibility or optimality bounds. This remains a limitation, especially for safety‑critical applications.

While feasibility is high, optimality gaps sometimes vary, and the method does not consistently outperform classical baselines on optimality.

**Reviewer Scores:**

Reviewer 3AaQ (initial 4): Likely would have increased to 6.

Reviewer nF3p (initial 6):  Likely would have maintained 6.

Reviewer WGVG (initial 6):  Likely would have maintained 6.

---

### Decision · Program_Chairs · 2026-01-26

Accept (Poster)